 

# Dynamics of the IFT machinery at the ciliary tip

**Alexander Chien[1†], Sheng Min Shih[2†], Raqual Bower[3], Douglas Tritschler[3], Mary E Porter[3], Ahmet Yildiz[1,2,4]\***

[1]Biophysics Graduate Group, University of California, Berkeley, Berkeley, United States; [2]Physics Department, University of California, Berkeley, Berkeley, United States; [3]Department of Genetics, Cell Biology and Development, University of Minnesota, Minneapolis, United States; [4]Department of Molecular and Cell Biology, University of California, Berkeley, Berkeley, United States

**Abstract** Intraflagellar transport (IFT) is essential for the elongation and maintenance of eukaryotic cilia and flagella. Due to the traffic jam of multiple trains at the ciliary tip, how IFT trains are remodeled in these turnaround zones cannot be determined by conventional imaging. Using PhotoGate, we visualized the full range of movement of single IFT trains and motors in *Chlamydomonas* flagella. Anterograde trains split apart and IFT complexes mix with each other at the tip to assemble retrograde trains. Dynein-1b is carried to the tip by kinesin-II as inactive cargo on anterograde trains. Unlike dynein-1b, kinesin-II detaches from IFT trains at the tip and diffuses in flagella. As the flagellum grows longer, diffusion delays return of kinesin-II to the basal body, depleting kinesin-II available for anterograde transport. Our results suggest that dissociation of kinesin-II from IFT trains serves as a negative feedback mechanism that facilitates flagellar length control in *Chlamydomonas*.

DOI: https://doi.org/10.7554/eLife.28606.001

**\*For correspondence:** yildiz@ berkeley.edu

[†]These authors contributed equally to this work

**Competing interests:** The authors declare that no competing interests exist.

## Introduction

Cilia (or eukaryotic flagella, terms essentially referring to the same organelle) are hair-like organelles that extend from the plasma membrane of nearly all mammalian cells. The core structural component of a cilium is the axoneme, a ring of nine unipolar doublet microtubules surrounding a central pair of singlet microtubules. Cilia play essential roles in cell motility, generate the movement of fluids over multiciliated surfaces, and sense extracellular signals (*Satir et al., 2010*). To assemble and maintain functional cilia, the IFT machinery (*Kozminski et al., 1993*) transports axonemal precursors and sensory proteins bidirectionally between the cell body and the ciliary tip. Defects in IFT are linked to a wide range of human diseases including Bardet-Biedl syndrome, retinal degeneration, and polycystic kidney disease (*Brown and Witman, 2014*).

Intraflagellar cargoes interact with multiprotein complexes known as IFT particles that are organized into larger IFT trains as they enter the flagellum (*Cole et al., 1998*; *Lechtreck et al., 2017*; *Pigino et al., 2009*). In most species, the IFT trains are transported anterogradely from the base to the tip of a flagellum by heterotrimeric kinesin-II (*Kozminski et al., 1995*), but some species also use a second homodimeric kinesin to build more specialized sensory cilia (*Snow et al., 2004*). Once the trains reach the tip, they are reorganized and transported retrogradely to the ciliary base by dynein-1b (*Pazour et al., 1999*; *Porter et al., 1999*; *Signor et al., 1999*). Along the length of a cilium, the activities of kinesin and dynein motors are reciprocally coordinated, such that only one type of a motor is active at a time (*Shih et al., 2013*). As a result, trains move between the tip and base of the cilium without significant pauses or back-and-forth motion (*Dentler, 2005*; *Engel et al., 2009*) and switch directions at the turnaround zones (*Ishikawa and Marshall, 2011*; *Laib et al., 2009*;

**eLife digest** Cilia and flagella are hair-like structures that protrude from nearly every human cell and play a number of roles including transmitting signals and enabling cells to move. These structures lengthen when new material is deposited at their tip by a process called intraflagellar transport (IFT). In this process, protein complexes known as IFT trains carry cargo along tracks that run along the length of each flagellum. Different motor proteins power the IFT trains in different directions: kinesin moves IFT trains from the base to the tip, while dynein moves them back in the opposite direction.

When IFT trains arrive at the tip of the flagellum, they release their cargo and undergo a major reorganization process in which the trains switch motors in order to move back to the base. Because the many IFT trains at the tip form a kind of 'traffic jam', standard imaging techniques have been unable to distinguish exactly what happens during this reorganization.

A new imaging method called PhotoGate microscopy can track individual molecules inside crowded cells. Chien, Shih et al. have now used this method to visualize the full range of movements made by IFT trains and motors in the flagella of a species of single-celled algae. This revealed that at the tip of the flagellum, IFT trains split apart and mix with each other to assemble into new trains, which move back to the base. In addition, kinesin carries dynein to the tip as inactive cargo, detaches from IFT trains at the tip and diffuses back to the base of the flagellum. This delays kinesin's return and causes it to accumulate in the flagellum, which helps to explain why flagella assemble more slowly as they lengthen: as the flagellum grows longer, less kinesin is available at the base to transport material to the tip. Thus kinesin diffusion helps the algae to regulate the length of their flagella.

Further research is now needed to study whether similar mechanisms control the length of flagella in other organisms. Defects in the intraflagellar transport process have been linked to a range of human diseases known as ciliopathies, and so the results presented by Chien, Shih et al. could also help us to uncover the causes and potential treatments for these conditions.
DOI: https://doi.org/10.7554/eLife.28606.002

*Shih et al., 2013*). Dynein-1b requires kinesin-II activity to reach the ciliary tip, suggesting that it travels on anterograde trains (*Iomini et al., 2001*; *Pedersen et al., 2006*; *Rompolas et al., 2007*). Retrograde traces of kinesin-II have not been frequently observed in *Chlamydomonas*, and it remains unclear how kinesin-II returns to the basal body (*Engel et al., 2009*; *Wingfield et al., 2017*). Because anterograde and retrograde IFT trains have different sizes and depart at different frequencies (*Dentler, 2005*; *Iomini et al., 2001*), IFT trains must be remodeled at the distal tip and the flagellar base.

The mechanism underlying the remodeling of IFT complexes at the ciliary tip and base cannot be directly observed by conventional microscopy methods because multiple trains coexist in these turn-around zones (*Buisson et al., 2013*; *Iomini et al., 2001*; *Wren et al., 2013*). In this work, we adapted PhotoGate (*Belyy et al., 2017*) to control the number of fluorescent IFT trains entering the flagellum of the unicellular algae *Chlamydomonas reinhardtii*. Using this approach, we monitored the turnaround behavior and remodeling of single IFT trains at the flagellar tip. We also elucidated the mechanisms by which the kinesin and dynein motors are recycled in this process and IFT trains reverse their direction of motion. The dynamics of IFT motor turnover at the tip suggest a new mechanism for how *Chlamydomonas* controls the length of its flagella.

## Results

### IFT trains split apart and mix with each other at the flagellar tip

To monitor IFT movement, we tracked the dynamic behavior of IFT27, a core component of the IFT complex B, in a *pf18 IFT27-GFP* strain (*Engel et al., 2009*; *Qin et al., 2007*). This strain has paralyzed flagella (*pf*) that readily adhere to the glass surface, enabling us to monitor IFT under total internal reflection (TIR) illumination (*Engel et al., 2009*). Consistent with previous studies (*Dentler, 2005*; *Engel et al., 2009*; *Iomini et al., 2001*; *Kozminski et al., 1993*; *Qin et al., 2007*; *Shih et al., 2013*), IFT trains moved processively along the length of flagella, reversing direction at

**Figure 1.** IFT trains split apart and mix with each other at the flagellar tip. (**a**) Kymograph of a surface-immobilized *pf18 IFT27-GFP* strain shows that IFT trains move bidirectionally along the flagellum, only reversing direction at the tip and the base. Multiple IFT trains accumulate at the flagellar tip. Representative anterograde and retrograde trajectories are shown with yellow and red dashed lines, respectively. (**b**) Schematic representation of the PhotoGate assay. (1) The distal half of the flagellum is prebleached by moving the powerful gate beam from the flagellar tip to near the base of the flagellum. (2–3) The gate beam is turned off to allow a single anterograde train to enter the flagellum without photobleaching. (4) The beam is then repeatedly turned on for 0.2 s to photobleach the successive trains entering the flagellum and (5) turned off for 0.8 s to image the single fluorescent

*Figure 1 continued on next page*

*Figure 1 continued*

train within the flagellum. Photobleached trains are not shown. (**c–e**) Kymographs of one (**c**), two (**d**) and three (**e**) fluorescent anterograde trains entering the flagellum. Anterograde trains pause at the flagellar tip and split into multiple retrograde trains that move back to the base. Arrival of fluorescent anterograde trains and departure of retrograde trains at the tip are shown with red and yellow stars, respectively. Arrows represent repetitive bleaching events near the base of the flagellum. (**f**) (Left) The number of fluorescent retrograde trains was quantified as a function of one, two or three fluorescent anterograde trains entering the flagellum after photobleaching. (Right) The average number of retrograde trains increased sub-proportionally with the number of fluorescent anterograde trains entering the flagellum. N = 97, 60, 42 train events from top to bottom, in 160 cells, from 13 independent experiments. (**g**) The number of detectable retrograde trains versus the numbers of incoming anterograde trains in PhotoGate experiments and Monte Carlo simulations (PB: photobleaching). Solid lines represent the fit of data to the power law ($y = ax^n$). $n$ is less than one under each condition. Error bars represent s.e.m. (N = 10,000 for simulations).

DOI: https://doi.org/10.7554/eLife.28606.003

The following figure supplements are available for figure 1:

**Figure supplement 1.** Anterograde and retrograde velocities of epitope-tagged IFT27, KAP, and D1bLIC.

DOI: https://doi.org/10.7554/eLife.28606.004

**Figure supplement 2.** Additional examples for PhotoGate imaging of IFT27, D1bLIC, and KAP.

DOI: https://doi.org/10.7554/eLife.28606.005

**Figure supplement 3.** Monte Carlo simulations for the dynamics of IFT trains at the flagellar tip.

DOI: https://doi.org/10.7554/eLife.28606.006

the flagellar tip and base (*Figure 1a*, *Video 1*). Pausing and reversals of anterograde trains before reaching the tip were very rare. The velocity of IFT27-GFP was 2.1 ± 0.4 µm s$^{-1}$ in the anterograde direction and 3.0 ± 0.7 µm s$^{-1}$ in the retrograde direction (*Figure 1—figure supplement 1*, mean ± s.d., N = 80 trains in each direction). Because a large number of GFP-labeled trains accumulated at the tip, the dwell and departure of individual trains at the tip could not be resolved by conventional TIR imaging (*Figure 1a*).

To monitor the turnaround behavior of individual IFT trains at the flagellar tip, we developed the one-dimensional PhotoGate assay (*Belyy et al., 2017*) to track single fluorescent complexes at the flagellar tip. In this assay, fluorescent trains located at distal parts of a flagellum were initially photobleached by moving a focused laser beam from the tip of the flagellum to near its base. We next opened the 'gate' by turning off the focused beam until a single fluorescent train entered the flagellum. The gate beam was repeatedly turned on for 0.2 s at 1 Hz to photobleach any additional anterograde trains entering the flagellum (*Figure 1b*, *Video 2*). Under these conditions, less than 1% of anterograde IFT trains were able to pass the gate unbleached. This approach revealed the full range of movement of single fluorescent IFT trains within the flagellum. IFT movement can be divided into three stages: anterograde movement toward the tip, pausing at the tip, and returning to the base by retrograde transport.

We directly observed that a single anterograde train splits into multiple retrograde trains at the tip (*Figure 1c–e*, *Figure 1—figure supplement 2a*). On average, 2.4 retrograde trains were detected departing from the tip after the arrival of a single fluorescent anterograde train (*Figure 1f*, N = 97), consistent with higher frequencies of retrograde IFT trains than anterograde IFT trains (*Dentler, 2005*; *Iomini et al., 2001*; *Qin et al., 2007*). However, the number of retrograde trains per fluorescent anterograde

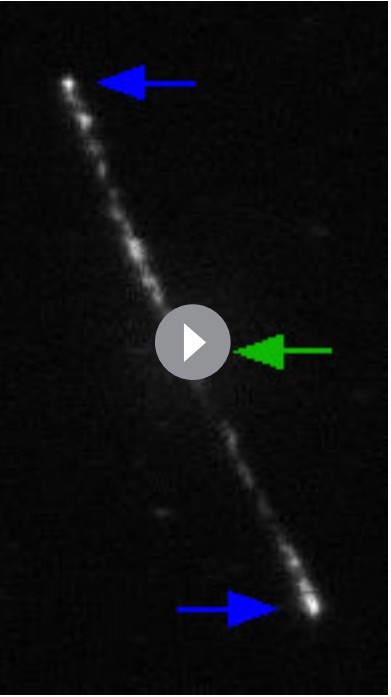

**Video 1.** Tracking of individual IFT trains in *Chlamydomonas*.

DOI: https://doi.org/10.7554/eLife.28606.007

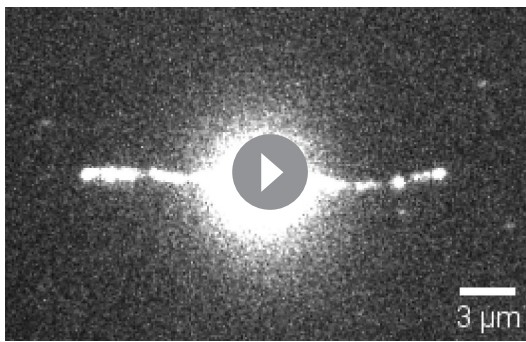

**Video 2.** Observing the dynamics of single IFT trains at the flagellar tip using the PhotoGate.
DOI: https://doi.org/10.7554/eLife.28606.008

**Video 3.** Tip return dynamics of two fluorescent anterograde trains using the PhotoGate.
DOI: https://doi.org/10.7554/eLife.28606.009

train in PhotoGate assays (2.4, *Figure 1c*) was significantly higher (Welch's t-test, p=10$^{-20}$) than the ratio of retrograde to anterograde train frequencies (1.15, *Figure 1a*) (*Dentler, 2005*; *Iomini et al., 2001*; *Reck et al., 2016*). These observations suggested that IFT complexes from different anterograde trains recombine with each other to form retrograde trains at the tip. To test this possibility, we closed the gate after two or three fluorescent anterograde trains entered the flagellum (*Figure 1d,e*, *Videos 3* and *4*) and measured the number and return frequency of retrograde trains departing from the tip. If individual trains split and return without mixing with each other, the number and frequency of fluorescent retrograde trains would be proportional to the number of fluorescent anterograde trains. In contrast, we observed 2.4, 3.6, and 4.2 returning trains on average for one, two, and three incoming trains, respectively (N = 97, 60, 42, *Figure 1f*). The return frequencies for one, two, and three incoming fluorescent trains were 0.57, 0.71, and 0.76 s$^{-1}$, respectively. Because the increase was sub-proportional with the number of anterograde trains, we concluded that the fluorescent complexes in the anterograde trains disassemble and mix with a pool of 'dark' complexes from the other photobleached trains at the tip before they reorganize into retrograde trains (*Figure 1f*). Monte Carlo simulations revealed that our conclusions are not markedly affected by the limited number of IFT27-GFPs per anterograde train (~6) or GFP photobleaching under TIR illumination (0.07 s$^{-1}$, *Figure 1g*, *Figure 1—figure supplement 3*, see Materials and methods).

## IFT tip turnaround is regulated by dynein activity and extracellular Ca$^{2+}$

To understand how IFT trains are processed at the tip, we analyzed the time between the arrival of an anterograde train and the departure of fluorescent retrograde trains (referred to as tip resting time) at the tip (*Figure 2a*). When only a single fluorescent anterograde IFT27-GFP train was left unbleached near the base of the flagellum, the tip resting time of the first retrograde train was 3.1 ± 0.3 s (mean ± s.e.m., N = 97, *Figure 2b*), comparable to that of IFT cargos (*Craft et al., 2015*; *Reck et al., 2016*; *Wren et al., 2013*). Tip resting time was independent of flagellar length (*Figure 2—figure supplement 1*). If IFT tip turnaround was rate-limited by a single process, we would expect a single exponential distribution of tip resting times. However, the tip resting time histogram of first retrograde trains fit well to a Gamma distribution with a shape parameter of 3 and a rate constant of ~1 s$^{-1}$, indicating that tip turnaround of IFT trains occurs rapidly through a multistep process (*Figure 2b*). We also observed that the tip resting time of first, second, third, and fourth fluorescent retrograde trains increases linearly (*Figure 2c*). Because the average time between successive retrograde trains (Δt = 1.7 s) is the same, we concluded that the tip departure is a purely stochastic process. The linear fit to the tip resting times has a y-intercept of 1.3 s (*Figure 2c*), revealing that the departure of the first train takes longer than Δt. Therefore, the complexes dwell at the tip through another rate limiting process before they can depart from the tip.

When two or three fluorescent anterograde trains were allowed to pass through the gate, Δt became shorter (1.4 and 1.3 s, Welch's t-test, p=0.03 and 0.02 for two and three trains, respectively; *Figure 2c*). This is because providing a higher number of fluorescent GFPs available at the tip

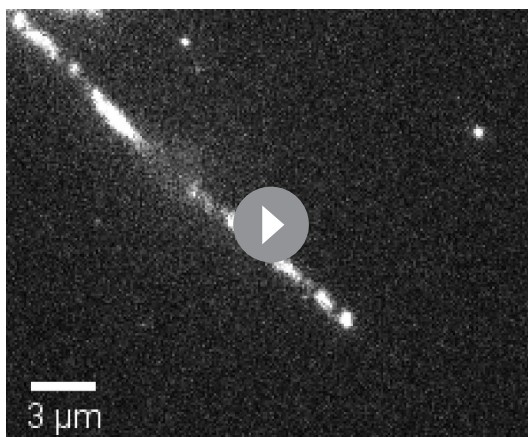

**Video 4.** Tip return dynamics of three fluorescent anterograde trains using the PhotoGate.
DOI: https://doi.org/10.7554/eLife.28606.010

increases the likelihood of retrograde trains to have at least one fluorescent GFP. Remarkably, the y-intercept remained constant when we allowed one to three more fluorescent antero-grade trains to enter the flagellum (*Figure 2c*), suggesting that this duration corresponds to processing and breakdown of anterograde trains at the tip (referred to as tip remodeling time). The time between tip remodeling and departure of IFT trains is defined as the tip departure time (*Figure 2a*).

We also tested whether extracellular calcium and the dynein inhibitor ciliobrevin D (*Firestone et al., 2012*) affect the duration of IFT tip turnaround. Calcium regulates pausing of IFT trains along the flagellum (*Collingridge et al., 2013*; *Shih et al., 2013*), and disrupting calcium-dependent kinesin-II phosphorylation causes abnormal accumulations of IFT proteins at the cil-iary tip (*Liang et al., 2014*). When extracellular calcium in media (0.34 mM) was chelated using 0.5 mM EGTA, the tip departure time increased to 2.8 s (Welch's t-test, p=0.01), whereas tip remod-eling time (1.4 s) remained unaltered (*Figure 2d,e*). Therefore, calcium has minimal effect on the breakdown of anterograde trains, but may have a regulatory role in the assembly or departure of retrograde trains. Addition of 0.1 mM ciliobrevin D to media results in 50% reduction in the fre-quency of retrograde and anterograde trains, and 50% and 28% reduction in retrograde and antero-grade train velocities, respectively (*Shih et al., 2013*). In this case, the tip resting time of IFT27 increased over two-fold (*Figure 2e*, $p<10^{-4}$), suggesting that rapid turnover of IFT trains depends on dynein activity.

## Kinesin-II dissociates from IFT trains at the tip

We next turned our attention to the movement of the IFT motors and their exchange at the flagellar tip. Dynein-1b was tagged with GFP at its light intermediate chain (D1bLIC), which assembles into the dynein-1b complex and rescues *d1blic* mutant phenotypes (*Reck et al., 2016*). In the *d1blic:: D1bLIC-GFP* strain, D1bLIC moved continuously in the anterograde and retrograde directions at velocities similar to that of the IFT trains (*Reck et al., 2016*) (*Figure 3a*, *Figure 1—figure supple-ment 1*). Kinesin-II was tagged with GFP at its non-motor subunit KAP that localizes kinesin-II to the flagellar base (*Mueller et al., 2005*). In the *fla3::KAP-GFP* strain, KAP moved primarily in the antero-grade direction to the flagellar tip at a similar speed to anterograde IFT27 (*Figure 3b*, *Figure 1—figure supplement 1*). Unlike D1bLIC, retrograde traces of KAP were not frequently observed (*Engel et al., 2009*; *Wingfield et al., 2017*), suggesting that kinesin-II dissociates from IFT trains at the tip (*Engel et al., 2009*; *Engel et al., 2012*).

PhotoGate assays revealed that D1bLIC-GFP has similar tip turnaround dynamics to IFT27-GFP (*Figure 3c*, *Figure 1—figure supplement 2b*, *Video 5*). After arrival of a single anterograde D1bLIC-GFP train at the tip, we detected on average 2.5 retrograde D1bLIC trains. The average tip resting time until the departure of the first retrograde train was 1.8 ± 0.2 s (mean ± s.e.m., N = 60, *Figure 3d*), with additional ~1.1 ± 0.1 s between subsequent departure events (*Figure 3e*). Photo-Gate imaging of KAP-GFP cells showed that single KAP-GFP trains moved anterogradely to the tip and rested at the tip for 2.2 ± 0.2 s (N = 95). Unlike D1bLIC, individual KAP-GFP particles moved away from the tip by rapid saltatory motion (*Figure 3f,g*, *Figure 1—figure supplement 2c*, *Video 6*). Mean square displacement (MSD) analysis showed that KAP undergoes one-dimensional diffusion at 1.68 ± 0.04 $\mu m^2$ $s^{-1}$ (mean ± s.e.m., N = 27 traces) within the flagellum after it departs from the tip (*Figure 3h*), consistent with the values measured for tubulin and EB1 that undergo diffusion within the ciliary space (*Craft et al., 2015*; *Harris et al., 2016*). The tip resting time of KAP remained nearly constant at different steady-state flagellar lengths (*Figure 2—figure supplement 1*) and was shorter



**Figure 2.** Tip turnaround of IFT trains is a multistep process regulated by dynein activity and extracellular $Ca^{2+}$. (a) The schematic describes the definition of tip resting time, remodeling time, and departure time measured from the kymographs. Arrival of the first fluorescent anterograde train and the departure of retrograde trains are shown with red and yellow stars, respectively. Tip resting time and departure time are only shown for the first retrograde train. Tip remodeling time is assumed to be the same for each train. Δt represents the time between the departure of successive retrograde trains. (b) The tip resting time histogram of the first retrograde IFT27-GFP train (dark grey) and all of the trains (light grey) emanating from a single anterograde train. The histogram of the first retrograde trains was fitted to a Gamma function (red curve). α and λ are shape and rate parameters, respectively. (c) The linear fit to the average tip resting time reveals Δt between successive trains. The y-intercept (black dashed line) represents the tip remodeling time. Errors represent standard error of the linear fit. (d) Average tip resting times of the first, second, third, and fourth retrograde IFT27-GFP trains coming out of an anterograde train for cells in TAP media (red, N = 97) and calcium-depleted media (blue, N = 44). Errors represent standard error of the linear fit. (e) IFT27-GFP tip resting times in 0.5 mM EGTA and 0.1 mM ciliobrevin D treated cells. The line within the boxplot represents the mean. The outer edges of the box represent standard deviation. N = 97, 241, 44, 109, 34, and 48 retrograde trains from left to right, in a total of 20 independent experiments (Welch's t-test, *p<0.05, ***p<0.001, as compared to no treatment for first retrograde trains).

DOI: https://doi.org/10.7554/eLife.28606.011

The following figure supplement is available for figure 2:

**Figure supplement 1.** Tip resting time is independent of flagellar length.

DOI: https://doi.org/10.7554/eLife.28606.012

than that of IFT27, indicating that departure of kinesin from the tip is independent of flagellar length and departure of retrograde trains (*Figure 2—figure supplement 1*).

Unlike IFT trains, the majority (89%, N = 95) of KAP-GFPs simultaneously departed from the tip in a single step (*Figure 3—figure supplement 1*). Because each anterograde train contains multiple (6) KAP-GFPs on average (*Engel et al., 2009*), this observation indicates that kinesin-II departs from the tip in the same oligomeric state as it arrives. 34% of kymographs clearly showed a single diffusing fluorescent spot after departure (*Figure 3f*), suggesting that KAPs are held together in a single complex. In 30% of kymographs, the KAP signal spread quickly along the length of a flagellum after departure, suggesting that kinesin-IIs can also diffuse alone (*Figure 3—figure supplement 2*,

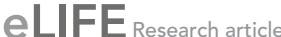

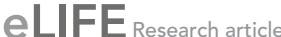

**Figure 3.** PhotoGate reveals the tip turnaround behavior of IFT motors. (a) In a conventional TIR assay, anterograde and retrograde D1bLIC-GFP traces were clearly visible, but the tip behavior of individual trains could not be discerned. (b) In a conventional TIR assay, KAP-GFP was observed to move anterogradely, but the retrograde transport of KAP was rarely observed. (c) PhotoGate imaging of D1bLIC-GFP shows that D1bLIC trains move to the tip anterogradely, split into multiple trains, and return to the base retrogradely. Red and yellow stars indicate arrival to and departure of D1bLIC-GFP from the tip, respectively. (d) The tip resting time histogram of D1bLIC-GFP. Tip resting time of the first retrograde trains are fit to a Gamma distribution (red curve; 95% c.i. for α is 1.30–2.52 and for λ is 0.69–1.47 s⁻¹). N = 60 anterograde trains in 60 cells over nine independent experiments. (e) Averaged tip resting time of the first, second, third, and fourth retrograde D1bLIC-GFP particles returning from the tip (mean ± s.e.m.). (f) Kymograph analysis of a KAP-GFP cell imaged by PhotoGate. KAP undergoes active transport in the anterograde direction, pauses at the flagellar tip, and diffuses back to the flagellar base. (g) The tip resting time histogram of KAP-GFP. The red curve represents a fit of first train resting times to a Gamma distribution. N = 95 anterograde trains in 47 cells over four independent experiments (95% c.i. for α is 1.87–3.19 and for λ is 0.88–1.60 s⁻¹). (h) MSD analysis of KAP-GFP diffusion after it leaves the flagellar tip reveals the average diffusion constant (N = 27, mean ± s.e.m.). (i) High-resolution tracking of a KAP-GFP particle reveals the two-dimensional trajectory during anterograde transport and diffusion. The red curve is the polynomial fit to the trace. (j) The residual plot to the trace in (i) reveals lateral fluctuations (σ) during anterograde transport (black) and diffusion (blue).

DOI: https://doi.org/10.7554/eLife.28606.013

The following figure supplements are available for figure 3:

**Figure supplement 1.** Tip resting time of KAP-GFP under EGTA and ciliobrevin D treatments.

DOI: https://doi.org/10.7554/eLife.28606.014

**Figure supplement 2.** Example kymograph of KAP particle breaking apart after tip departure.

DOI: https://doi.org/10.7554/eLife.28606.015

*Video 7*). The rest of the kymographs were ambiguous. Similar to IFT27, the tip resting time of KAP increased ~50% when the cells were treated with ciliobrevin D (p=0.0016), but EGTA had no significant effect on tip resting time of KAP (*Figure 3—figure supplement 1*).

We next investigated whether KAP slides linearly along the microtubule track, similar to the non-processive, microtubule-depolymerizing kinesin MCAK (*Helenius et al., 2006*). In this case, KAP

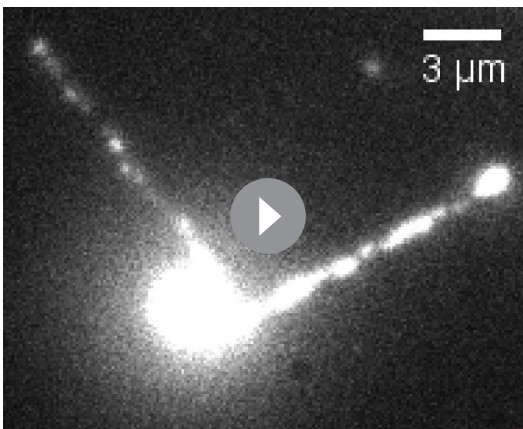

**Video 5.** Tip return dynamics of D1bLIC-GFP.
DOI: https://doi.org/10.7554/eLife.28606.016

clusters are expected to move along the microtubule long axis, so the fluctuation in KAP position in the perpendicular axis would be similar to the error of single particle tracking. The KAP-GFP particles had lateral fluctuations of $19 \pm 2$ nm (mean $\pm$ s.d.) when moving in the anterograde direction. After departing from the tip, lateral fluctuations of diffusing spots increased to $65 \pm 7$ nm (*Figure 3i,j*), comparable to the radius of the axoneme. The intensity of fluorescent spots stayed relatively constant during anterograde transport and diffusion, suggesting that the measured lateral fluctuations are due to diffusive motion rather than decreased tracking precision. We concluded that after KAP detaches from the flagellar tip, it freely explores the space between the flagellar membrane and the axonemal surface rather than sliding along microtubules.

## Kinesin-II carries dynein-1b as an inactive passenger during anterograde IFT

To investigate how kinesin-II and dynein-1b motors interact with anterograde and retrograde trains and how they are recycled back to the basal body, we transformed a *d1blic* mutant with both *D1bLIC-crCherry* and *KAP-GFP* constructs and simultaneously tracked the movement of KAP and D1bLIC subunits in the rescued cells (*Figure 4—figure supplement 1*, *Video 8*). The *D1bLIC-crCherry* transgene rescued the flagellar assembly defects in the *d1blic* mutant, increasing the flagellar length to $12.2 \pm 1.6$ μm (mean $\pm$ s.d., N = 100 flagella). Both tagged motors were expressed at near wild-type levels (*Figure 4—figure supplement 1*). The velocities of anterograde and retrograde D1bLIC-crCherry trains were similar to those observed with IFT27-GFP (*Figure 4a*, *Figure 1—figure supplement 1*). We observed strong co-localization of D1bLIC-crCherry and KAP-GFP on anterograde trajectories (*Figure 4a*), demonstrating that dynein-1b is carried to the flagellar tip by kinesin-II. Only D1bLIC-crCherry trains showed robust retrograde transport, while retrograde traces of KAP-GFP were rarely observed, consistent with dissociation of kinesin-II from IFT trains at the tip.

To determine which motor first departs from the tip after the arrival of an anterograde train, we performed two-color Photogate experiments to simultaneously track KAP-GFP and D1bLIC-crCherry from individual anterograde trains (*Figure 4b*). Out of 21 cells, KAP began diffusive motion before the retrograde movement of D1bLIC in 10 cells (*Figure 4b*), D1bLIC left the tip before KAP in 8 cells (*Figure 4—figure supplement 2*), and both appeared to exit the tip simultaneously (within 0.24 s) in 3 cells. These results suggest that kinesin-II and dynein-1b exit the flagellar tip independently from each other.

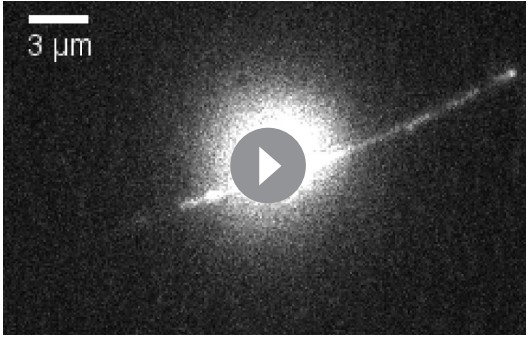

**Video 6.** KAP-GFP dissociates from IFT trains at the tip.
DOI: https://doi.org/10.7554/eLife.28606.017

## Kinesin-II returns from the ciliary tip to the cell body by diffusion

Dissociation of KAP from IFT trains at the tip is consistent with the recycling of kinesin-II to the cell body in the absence of active retrograde IFT (*Engel et al., 2012*; *Pedersen et al., 2006*; *Reck et al., 2016*). However, it remained unclear how kinesin-II achieves this long-range movement without active transport. To test whether diffusion from the tip effectively recycles KAP to the cell body, we performed fluorescence recovery after photobleaching (FRAP) assays in the middle sections of full-length flagella of *fla3*::KAP-GFP cells (~12 μm, *Figure 5a*, *Video 9*). Directional

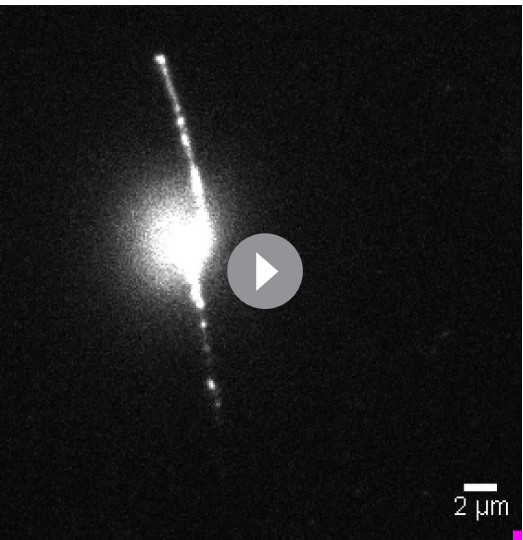

**Video 7.** Diffusing KAP-GFP particles can break apart after leaving the tip.
DOI: https://doi.org/10.7554/eLife.28606.018

movements of KAP-GFP labeled trains into the photobleached area were seen from the antero-grade direction, whereas recovery of GFP fluorescence from the retrograde direction was primarily due to diffusion of KAP-GFP from the tip. These results suggest that the high levels of background observed in KAP-GFP flagella were caused by kinesin-II motors dissociated from IFT trains at the tip. The diffusion constant calculated from the fluorescence recovery ($1.8 \pm 0.1$ μm$^2$ s$^{-1}$, *Figure 5b*) was similar to the result of the MSD analysis (*Figure 3h*). The fluorescent background in KAP-GFP flagella increased towards the tip, suggesting a net efflux of diffusing KAP-GFP towards the cell body (*Figure 5c,d*). During flagellar regrowth, the KAP-GFP gradient was maintained for all flagellar lengths (*Figure 5—figure supplement 1a*, see Materials and methods). The influx of KAP-GFP fluorescence to the flagellum through anterograde IFT was statistically indistinguishable from the efflux of KAP-GFP to the base through one-dimensional diffusion in flagella (Welch's t-test, p=0.80, N = 57, *Figure 5—figure supplement 2*, see Materials and methods). These results strongly indicate that KAP-GFP returns to the cell body by diffusing from the flagellar tip.

We ran Monte Carlo simulations to estimate the accumulation of KAP in a flagellum at a steady-state using the measured values of IFT train loading (*Engel et al., 2009*), diffusion coefficient, flagellar length, and IFT train frequency. The model assumes that KAP is released from anterograde IFT trains at the tip, diffuses within a flagellum, and is taken up by the basal body. Under these conditions, simulations confirmed the build-up of a linear concentration gradient of KAP in the flagellum (*Figure 5—figure supplement 1b*). In fully-grown flagella, the return of KAP to the flagellar base takes 42 s on average, an order of magnitude longer than the travel of retrograde trains (4 s) to the base. This delay leads to a ~4 fold higher amount of KAP inside the flagellum compared to a case in which KAP returns to the base with retrograde trains (*Figure 5—figure supplement 1c*). Unlike KAP, IFT27-GFP cells have a low fluorescence background without an obvious concentration gradient along the length of the flagellum (*Figure 5d*) due to active transport of the IFT trains in both directions.

## Kinesin-II is depleted from the basal body during flagellar regrowth

KAP-GFP loading on IFT particles has been shown to decrease with increasing flagellar length (*Engel et al., 2009*), but the underlying mechanism remained unclear. Because a larger amount of KAP builds up in the flagellum as the flagella elongate (*Figure 5—figure supplement 1*), loading of KAP onto the subsequent IFT trains may be reduced by depletion of KAP at the flagellar base. To test this model, we deflagellated *fla3::*KAP-GFP cells and measured the GFP fluorescence at the basal body and in the flagellum during flagellar regrowth using confocal microscopy (*Figure 6a*). The total amount of KAP localized to the base and flagellum increased by two-fold with flagellar length, indicating the upregulation of IFT components during flagellar growth. The fluorescence intensity at the flagellar base was highest for short flagella (1–4 μm) and decreased ~4 fold as cells grew full-length flagella (~10 μm, *Figure 6b*), significantly larger than ~1.6 fold reduction reported previously (*Ludington et al., 2015*). We also observed that the KAP fluorescence in the flagellum was low in short flagella and increased ~10 fold as the flagellar length reached the steady-state (*Figure 6b*).

Changes in the amount of IFT complexes were markedly different from that of KAP during flagellar regrowth (*Figure 6a*). Unlike KAP-GFP, basal body fluorescence of IFT20-GFP remained nearly constant across all flagellar lengths in IFT20::IFT20-GFP cells (*Figure 6c*), presumably because they



**Figure 4.** Transport roles of kinesin-II and dynein-1b. (**a**) Representative kymographs of KAP-GFP and D1bLIC-crCherry in a *d1blic::D1bLIC-crCherry KAP-GFP* flagellum. KAP-GFP and D1bLIC-crCherry co-localize on the IFT trains in the anterograde direction. Retrograde tracks are seen in the D1bLIC-crCherry channel, but are rarely visible in the KAP-GFP channel. (**b**) Example two-color PhotoGate trace of KAP-GFP (left) and D1bLIC-crCherry (middle) in single flagella. KAP-GFP and D1bLIC-crCherry arrive at the tip on the same train. In this example, KAP diffuses away from the tip before the departure of D1bLIC trains (right). Red and yellow stars indicate arrival to and departure from the flagellar tip, respectively. (**c**) A model for the turnover of IFT trains and motors at the flagellar tip. Kinesin-II motors transport individual anterograde IFT trains to the flagellar tip. Dynein-1b is carried with anterograde trains as an inactive passenger. At the tip, IFT complexes detach from microtubules, disassemble, and mix with the tip protein pool to assemble new trains. These trains are transported retrogradely by dynein-1b. Kinesin-II detaches from IFT trains at the flagellar tip and diffuses back to the base by diffusion either as a cluster (blue dashed circle) or individually.

*Figure 4 continued on next page*

*Figure 4 continued*

DOI: https://doi.org/10.7554/eLife.28606.019

The following figure supplements are available for figure 4:

**Figure supplement 1.** Expression of KAP-GFP and D1bLIC-crCherry in isolated flagella from a double-tagged strain.
DOI: https://doi.org/10.7554/eLife.28606.020

**Figure supplement 2.** Example two-color PhotoGate trace of KAP-GFP (left) and D1bLIC-crCherry (right) in single flagella.
DOI: https://doi.org/10.7554/eLife.28606.021

are rapidly returned to the base through active transport. We also observed an increase of the GFP signal in the flagellum with elongation (*Figure 6c*), in contrast to the previous observation that the total amount of IFT components remains constant during flagellar regeneration (*Marshall and Rosenbaum, 2001*). This discrepancy may be related to differences in methods for quantifying IFT components in flagella.

We next determined the localization of KAP-GFP to the basal body and flagellum in cells that grow abnormally long and short flagella. *Chlamydomonas* grows ~1.5X longer flagella in the presence of Li$^+$ (*Nakamura et al., 1987*) by recruiting flagellar proteins from the cell body pool into the flagella (*Nakamura et al., 1987*) rather than requiring new protein synthesis (*Wilson and Lefebvre, 2004*). Consistent with previous observations, the KAP-GFP strain grew longer flagella in 50 mM Li$^+$. After reaching the steady state length, we calculated the total KAP fluorescence at the basal body and the flagellum (*Figure 6d*, *Figure 6—figure supplement 1*). In agreement with our model, we observed that KAP gets depleted at the basal body at equilibrium. The KAP fluorescence localized to a flagellum correlated strongly with flagellar length (Pearson's R = 0.86), similar to untreated cells. The total KAP fluorescence in the flagellum was 50% higher than untreated cells.

In the absence of new protein synthesis, *Chlamydomonas* can grow half-length flagella after deflagellation, suggesting that the cytoplasmic pool of flagellar proteins is at least one half of that localized to the flagellar compartment (*Rosenbaum et al., 1969*). We deflagellated KAP-GFP cells with pH shock and regrew their flagella in the presence of the protein synthesis inhibitor cycloheximide. In agreement with untreated and Li$^+$ treated cells, we observed that the KAP fluorescence at the flagellum correlates strongly with the flagellar length (Pearson's R = 0.86), whereas the KAP intensity was depleted at the basal body (*Figure 6d*, *Figure 6—figure supplement 1*). Total KAP fluorescence was one half of untreated cells, consistent with the fact that a large amount of KAP is lost during deflagellation. Therefore, over a wide range of flagellar lengths (2–22 μm), KAP gets depleted at the basal body when the flagella reach their equilibrium length.

## Discussion

### Remodeling of IFT complexes and motors at the flagellar tip in *Chlamydomonas*

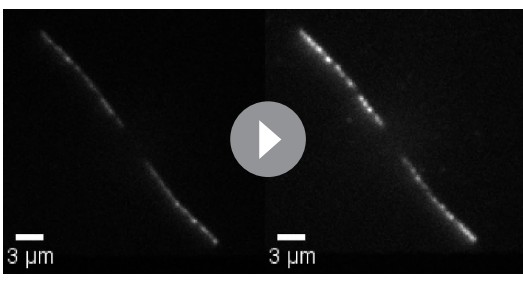

**Video 8.** Dual color imaging of KAP-GFP and D1bLIC-crCherry.
DOI: https://doi.org/10.7554/eLife.28606.022

Using PhotoGate, we have visualized the turn-around behavior of individual components of the IFT machinery at the flagellar tip. We present evidence that when IFT trains arrive at the tip, the complexes split apart and mix with complexes from other trains at the flagellar tip before initiating retrograde transport (*Figure 4c*). This dynamic disassembly and reassembly process may lead to differences in size, shape, and structure of anterograde and retrograde trains, as previously suggested by transmission electron microscopy (*Dentler, 2005*; *Stepanek and Pigino, 2016*). Remarkably, remodeling of IFT trains is completed within 1.3 s, with a 1.7 s average waiting time between the departures of successive trains, consistent with

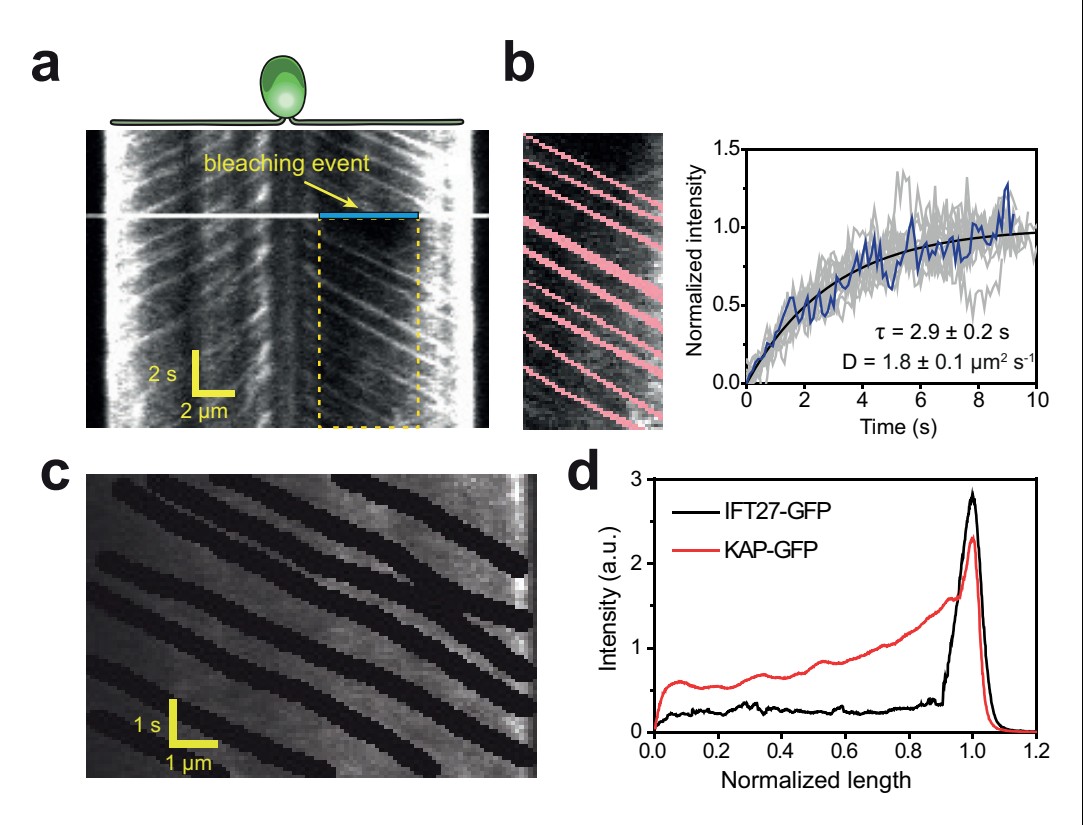

**Figure 5.** Diffusion of KAP from the flagellar tip leads to a concentration gradient along the flagellum. (**a**) Kymograph of KAP-GFP movement before and after photobleaching the middle section of the flagellum (blue area). While fluorescence recovery from the base is through anterograde movement, the recovery from the tip region is due to diffusion. (**b**) (Left) The GFP signal of anterograde traces (red) was manually subtracted from the rectangular area shown in (**a**). (Right) The intensity in the photobleached area shows recovery as a function of time (blue line). The average recovery signal of 13 cells (grey lines) was fitted to a one-dimensional diffusion equation (black curve,±95% c.i.). (**c**) In conventional TIR imaging, anterograde trajectories of KAP-GFP were manually subtracted from the kymograph. (**d**) The average GFP signal along the length of a flagellum in KAP-GFP and IFT27-GFP cells after the removal of anterograde and retrograde transport traces from the kymographs. Flagellar base and tip positions were normalized to 0 and 1, respectively. N = 11 for both KAP-GFP and IFT27-GFP.
DOI: https://doi.org/10.7554/eLife.28606.023

The following figure supplements are available for figure 5:

**Figure supplement 1.** A gradient of KAP-GFP fluorescence along the length of the flagellum exists across all flagellar lengths.
DOI: https://doi.org/10.7554/eLife.28606.024

**Figure supplement 2.** The influx and efflux of KAP-GFP fluorescence in fully grown flagella are equal.
DOI: https://doi.org/10.7554/eLife.28606.025

previously measured values for IFT complex subunits, dynein-1b, and other axonemal cargoes (*Craft et al., 2015*; *Qin et al., 2007*; *Reck et al., 2016*; *Wren et al., 2013*). Kinetic analysis of the tip resting time revealed that disassembly of anterograde trains and reassembly of the retrograde trains is a multistep process regulated by extracellular calcium and the concentration of active dynein motors.

Simultaneous tracking of the anterograde motor kinesin-II (with KAP-GFP) and the retrograde motor dynein-1b (with D1bLIC-Cherry) has further revealed significant differences in how these motors are recycled back and forth within a flagellum. Kinesin-II drives anterograde trains in a highly processive fashion and then dissociates from the IFT trains when they reach at the flagellar tip. KAP-GFP then returns to the flagellar base by diffusing within the flagellum, similar to the diffusion of kinesin-1 in mammalian neurons (*Blasius et al., 2013*) and in contrast to the retrograde transport of

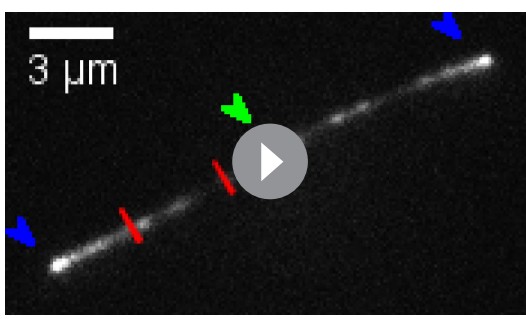

**Video 9.** Recovery of KAP-GFP after photobleaching the middle section of a flagellum.
DOI: https://doi.org/10.7554/eLife.28606.026

kinesin-II observed in other cilia (*Signor et al., 1999*; *Broekhuis et al., 2014*; *Williams et al., 2014*; *Prevo et al., 2015*), see below). We propose that the diffusion of KAP-GFP represents the movement of the entire heterotrimeric kinesin-II complex because KAP and the kinesin-II motor subunits co-sediment in sucrose density gradients of purified flagella extracts (*Cole et al., 1998*; *Mueller et al., 2005*) and neither KAP nor FLA10 accumulate in flagella during inactivation of retrograde transport (*Engel et al., 2012*; *Pedersen et al., 2006*; *Reck et al., 2016*). In certain cases, KAP-GFP appears to diffuse in an oligomeric form. It remains to be studied what holds KAPs together and whether other components of the IFT trains diffuse with KAP clusters after splitting and mixing at the tip.

The retrograde motor dynein-1b is transported to the flagellar tip on anterograde trains (*Reck et al., 2016*). Because kinesin and dynein motors do not compete against each other in a tug-of-war on anterograde trains (*Shih et al., 2013*), we concluded that dynein-1b is carried as an inactive motor complex (*Toropova et al., 2017*; *Zhang et al., 2017*), and it actively engages with microtubules only when it reaches the flagellar tip (*Figure 4c*). The average tip resting time of dynein-1b is similar to kinesin-II (Welch's t test, p=0.05), but the initiation of retrograde transport by dynein-1b does not require departure of kinesin-II motors from the tip, suggesting that these processes are independent from each other.

## The dynamic behavior of the IFT motors in *Chlamydomonas* flagella differs from that in other cilia

Several studies have revealed differences in IFT in the cilia and flagella of different organisms (*Prevo et al., 2017*). First, the microtubule tracks can vary considerably. In *Chlamydomonas*, the axoneme contains nine doublet microtubules, each composed of a complete A-tubule (with 13 protofilaments) and an incomplete B-tubule (with 10 protofilaments). The B-tubule terminates before the A-tubule less than 1 µm from the flagellar tip (*Ringo, 1967*; *Satish Tammana et al., 2013*). Recent electron microscopy studies have revealed that anterograde IFT trains are transported primarily on the B-tubule, whereas retrograde IFT trains are transported on the A-tubule (*Stepanek and Pigino, 2016*). The pausing of IFT particles observed at the flagellar tip in *Chlamydomonas* may therefore reflect not only the time involved in the re-organization of the IFT particles, but also the time required for switching between microtubule tracks.

In other cilia, such as *C. elegans* sensory and mammalian olfactory cilia, the proximal doublet microtubule segment is shorter, and the distal singlet MTs can vary significantly in length (~2.5 µm in *C. elegans* to >100 µm in mouse olfactory cilia). These cilia also employ two different kinesin-II motors for anterograde IFT (*Prevo et al., 2015*; *Snow et al., 2004*; *Williams et al., 2014*). In *C. elegans*, heterotrimeric kinesin-II is concentrated near the basal body region and transports anterograde IFT particles into the proximal doublet segment, where they are gradually handed over to a homodimeric kinesin-II, OSM-3, for transport into the distal singlet segment (*Prevo et al., 2015*; *Snow et al., 2004*). Unlike *Chlamydomonas*, both the heterotrimeric and homodimeric kinesin-II motors are recycled to the ciliary base by retrograde IFT, not by diffusion, in these and other metazoan cilia studied to date (*Broekhuis et al., 2014*; *Mijalkovic et al., 2017*; *Prevo et al., 2015*; *Signor et al., 1999*; *Williams et al., 2014*).

Another important difference between IFT in *Chlamydomonas* and metazoan cilia is the dynamic behavior of the motors themselves. IFT particles and motors move processively in the *Chlamydomonas* flagellum (*Dentler, 2005*), with little or no evidence for the frequent pausing and reversal along the axoneme previously described in *C. elegans* or mouse olfactory sensory cilia (*Mijalkovic et al., 2017*; *Prevo et al., 2015*). We also did not observe acceleration and deceleration of IFT motors near turnaround zones, nor the instantaneous (<600 ms) reversal of dynein-1b at the ciliary tip described in *C. elegans* (*Mijalkovic et al., 2017*; *Prevo et al., 2015*). The reasons for these

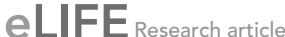

**Figure 6.** Kinesin-II accumulates in flagella and is depleted at the basal body during flagellar growth. (a) Representative confocal images show the distribution of IFT20-GFP and KAP-GFP fluorescence at the basal body region and in the flagella during flagellar regrowth. (b,c) Integrated KAP-GFP (b) and IFT20-GFP (c) fluorescence at the basal body (top), in the flagellum (middle), and in both regions (bottom) at different flagellar lengths. Each black dot represents a single flagellum and the blue line is the running average (± s.e.m.). For KAP-GFP, N = 104 flagella from 70 cells over two independent experiments. For IFT20-GFP, N = 103 flagella from 56 cells over two independent experiments. (d) Integrated KAP-GFP fluorescence at the basal body (left), in the flagellum (middle), and in both regions (right) in control cells was compared to cells treated with 50 mM Li$^+$, and cells that regrow their

*Figure 6 continued on next page*

*Figure 6 continued*

flagella after cycloheximide treatment. R represents Pearson's correlation coefficient. Each dot represents a single flagellum. For control cells, N = 66 flagella from 33 cells. For lithium-treated cells, N = 74 flagella from 37 cells. For cycloheximide-treated cells, N = 40 flagella from 20 cells. (e) A model for flagellar length control. When the flagellum is short, IFT trains contain more kinesin-II from the large basal body pool. As flagella elongate, the number of kinesin-II per IFT train decreases because a significant fraction of the kinesin-II unloads at the tip and undergoes diffusion in the flagellar lumen, depleting the kinesin-II pool at the flagellar base. (f) An analogy for kinesin-II loading on IFT trains. Passengers (dynein-1b) travel from the left shore (basal body) to the right shore (flagellar tip) on a boat (IFT trains) with oars (kinesin-II). At the right shore, the passengers get out and walk the boats back to the dock. Oars are left on the water and can only be collected when they randomly float back to the left shore. If the distance between the shores is large, oars build up on the water and are not readily available for new boats at the left shore.

DOI: https://doi.org/10.7554/eLife.28606.027

The following figure supplement is available for figure 6:

**Figure supplement 1.** Length and total fluorescence distributions of lithium and cycloheximide-treated cells.

DOI: https://doi.org/10.7554/eLife.28606.028

differences in IFT dynamics and turnover remain unknown, but they may be related to the variations in ciliary structure and organization described above, differential phosphorylation of kinesin-II motors (*Liang et al., 2014*), posttranslational modification of the microtubule tracks (*Stepanek and Pigino, 2016*), and other unidentified factors. In addition, both the frequency and speed of IFT is higher in *Chlamydomonas* (*Dentler, 2005*; *Engel et al., 2009*; *Engel et al., 2012*; *Iomini et al., 2001*; *Reck et al., 2016*; *Snow et al., 2004*; *Williams et al., 2014*; *Wingfield et al., 2017*) than that measured thus far in metazoan cilia (*Li et al., 2015*; *Yi et al., 2017*). This may allow *Chlamydomonas* to rapidly adjust the length of its flagella in response to internal or external stimuli, whereas most cilia in *C. elegans* sensory neurons or mammalian cells do not undergo extensive structural rearrangements once formed.

## A model for flagellar length control in *Chlamydomonas*

Cilia and flagella serve as a model system to study how cells precisely control organelle size because they elongate only in one direction. According to the balance point model, flagellar length is set when flagellar assembly and disassembly rates reach equilibrium (*Marshall et al., 2005*). While the disassembly rate is independent of flagellar length (*Kozminski et al., 1995*), the assembly rate is determined by the injection of IFT trains (*Marshall et al., 2005*). The amount of material being transported by these trains to the tip is correlated strongly with the amount of material localized to the flagellar base (*Ludington et al., 2013*; *Ludington et al., 2015*; *Wren et al., 2013*), which serves as a loading dock. Previous studies showed that IFT train size and the number of ciliary cargos per train scales inversely with flagellar length (*Craft et al., 2015*; *Engel et al., 2009*), consistent with the reduction of the assembly rate as flagella elongate (*Marshall et al., 2005*). However, it remained unclear which essential component of the IFT machinery limits the assembly of IFT trains at the basal body during elongation.

We propose that dissociation of kinesin-II from IFT trains serves as a negative feedback mechanism to control the assembly rate in *Chlamydomonas*. Our results show that the majority of kinesin-II dissociates from IFT trains at the flagellar tip and diffuses within the flagellum. Diffusion leads to a large accumulation of kinesin-II in the flagellum as the flagellum grows longer, while the amount of kinesin-II at the base decreases several-fold. As a result, lower amounts of kinesin-II are available to bring new anterograde IFT trains to the flagellar tip. This may lead to a reduction in the IFT train size and the rate of flagellar assembly as the flagella elongate (*Figure 6e,f*). Indeed, a recent theoretical study demonstrated that increased diffusion time of kinesin-II in longer flagella can explain the inverse relationship between length and IFT motor recruitment rate (*Hendel et al., 2017*).

Consistent with this model, previous studies showed that KAP intensity at the basal body correlates with KAP loading on IFT trains and the assembly rate during flagellar regeneration (*Ludington et al., 2013*). In the temperature-sensitive mutant strain *fla10^ts*, inactivation of kinesin-II motility ceases IFT and leads to resorption of the flagellum at a constant rate (*Kozminski et al., 1995*; *Marshall et al., 2005*). At intermediate temperatures, flagellar length correlates strongly with the estimated fraction of active kinesin-II motors in *fla10^ts* cells (*Marshall and Rosenbaum, 2001*), indicating that the amount of active kinesin-II limits flagellar growth. The experiments performed under Li$^+$ and cycloheximide treatments (*Figure 6d*) also support the idea that altering the amount

of KAP available for the flagellar compartment positively correlates with the flagellar length and that the equilibrium length is set when KAP gets depleted below a certain threshold at the basal body. We note that the KAP intensity at the flagellar base was lower (34–40%) in Li$^+$ and cycloheximide treated cells than in control cells at equilibrium (p<0.0001, *Figure 6d*, left). Although the reason for this difference remains unclear, it could be due to a reduction of KAP concentration in the cytoplasm or changes in the flagellar disassembly rate under treatment (*Wilson and Lefebvre, 2004*).

Unlike kinesin-II, IFT components are rapidly recycled to the cell body by dynein-1b and the amount of these components at the flagellar base remains nearly constant as the flagella elongate. Therefore, the abundance of IFT components and dynein-1b at the flagellar base is not limiting to maintain the length, consistent with previous observations that a small amount of IFT complexes and dynein-1b is sufficient to maintain fully grown flagella (*Reck et al., 2016*; *Wang et al., 2009*).

Diffusion is also proposed to play a role in setting the length of bacterial flagella (*Renault et al., 2017*), long polymers made from a single protein flagellin. Similar to the flagellar length control model originally proposed for *Chlamydomonas* (*Levy, 1974*), flagellins are injected into the channel of the filament and they diffuse to reach the assembly site at the filament tip, generating a concentration gradient decreasing towards the tip. As the filament elongates, it grows more slowly because it takes longer for the components to reach the tip. In contrast to bacterial flagellin, structural components are carried to the tip by IFT in eukaryotic flagella. In *Chlamydomonas*, we showed that diffusion of kinesin-II from the tip sets a concentration gradient decreasing towards the basal body, and its return to the flagellar base is delayed as the flagella elongate. This delay limits the amount of kinesin-II available for building longer flagella.

Our model is challenged by studies showing that the kinesin-II mutant strains *fla10$^{ts}$* and *fla3* maintain nearly full-length flagella at permissive temperatures although they accumulate significantly lower amounts of kinesin-II in the flagellar compartment (*Kozminski et al., 1995*; *Mueller et al., 2005*; *Pedersen et al., 2006*). Remarkably, *fla3* cells exhibit slower flagellar regeneration (*Mueller et al., 2005*), consistent with our prediction that the lower amount of kinesin-II negatively affects the assembly rate. However, a more recent study showed that *fla10$^{ts}$* flagella contain wild-type levels of kinesin-II at permissive temperatures (*Wang et al., 2009*). Given these apparent discrepancies, more quantitative approaches will be required to address whether the amount of kinesin-II correlates with flagellar length.

According to the balance-point model, flagella that contain lower amounts of kinesin-II can still maintain nearly full-length if they also have a lower disassembly rate. The studies that reported a reduction in kinesin-II expression in *fla10$^{ts}$* and *fla3* cells also noted significantly reduced anterograde IFT frequency and IFT particle subunits in flagella (*Mueller et al., 2005*; *Pedersen et al., 2006*). This could negatively affect the disassembly rate because IFT is required for efficiently removing certain axonemal precursors (*Qin et al., 2004*) and resorbing the flagellum prior to mitosis. Indeed, flagellar resorption before mitosis occurs at a faster rate than flagellar disassembly after inactivation of IFT (*Marshall et al., 2005*; *Pan and Snell, 2005*).

The mechanisms that control the expression of IFT components after deflagellation, regulate the exchange of material between the basal body and cytoplasm, and load material onto IFT trains must also play a major role in determining the length of flagella. Several studies have shown that IFT components are upregulated and accumulate in large numbers at the flagellar base after deflagellation (*Albee et al., 2013*; *Lefebvre and Rosenbaum, 1986*; *Stolc et al., 2005*). Additionally, a large pool of IFT components in the cytoplasm partially exchanges with the flagellar pool (*Buisson et al., 2013*; *Engel et al., 2009*; *Wingfield et al., 2017*) because cells can grow half-length flagella after deflagellation under complete inhibition of protein synthesis (*Rosenbaum et al., 1969*). However, molecular cues that govern these processes remain poorly understood and further studies in mutant cell lines that have abnormally long (*Nguyen et al., 2005*; *Tam et al., 2007*) or short flagella may provide new insight for the mechanism of flagellar length control.

## Materials and methods

### Strains and cell culture

The *pf18 IFT27-GFP* strain was obtained from the Marshall laboratory (University of California San Francisco) after crossing the *IFT27-GFP* transgene into the *pf18* background as previously described

(*Engel et al., 2009*; *Qin et al., 2007*). The *ift20::IFT20-GFP* strain (*Lechtreck et al., 2009*) was obtained from the Lechtreck laboratory (University of Georgia). The *fla3::KAP-GFP* (*Mueller et al., 2005*) and *d1blic::D1bLIC-GFP* (*Reck et al., 2016*) strains are available from *Chlamydomona*s Resource Center at the University of Minnesota (RRID: SCR_014960, Minneapolis, MN). The *d1blic:: D1bLIC-crCherry KAP-GFP* strain was generated as described below. These strains were not authenticated or tested for mycoplasma contamination. Strains were maintained on plates of TAP media containing 1% agar. For light microscopy, vegetative cells were resuspended in liquid TAP media at 22°C for 24–48 hr and passaged to fresh liquid TAP before introduction into a flow chamber.

### Isolation and characterization of the *d1blic::D1bLIC-crCherry KAP-GFP* strain

The *D1bLIC-crCherry* construct was generated by subcloning a *Chlamydomonas* codon-optimized version of the Cherry tag into a genomic copy of the *D1bLIC* gene (*Reck et al., 2016*). The Cherry tag was amplified by PCR from the plasmid pBR9 mCherryCr (*Rasala et al., 2013*) and inserted into a unique *Asc*I site located in the last exon of *D1bLIC*. The D1bLIC-crCherry construct was linearized with *Bam*HI and co-transformed into *d1blic* (CC-4487) with the selectable marker pSI103 and plated on TAP medium plus 10 μg/ml paromomycin. 960 transformants were picked into TAP media and screened for changes in colony morphology. 84 colonies were further examined by both phase contrast and fluorescence microscopy for rescue of flagellar assembly and expression of Cherry. Isolated flagella from four colonies were analyzed by Western blot for the presence of full-length D1bLIC-Cherry. A single colony was selected for a second round of transformation using the *KAP-GFP* construct (*Mueller et al., 2005*) and the plasmid pHyg3 (*Berthold et al., 2002*) and selection on 10 μg/ ml of hygromycin B. Two out of 96 transformants were identified as positive for both GFP and Cherry by fluorescence microscopy, and western blots of isolated flagella confirmed the presence of both D1bLIC-Cherry and KAP-GFP in the rescued strains. Antibodies used included a rat antibody against *Chlamydomonas* KAP (*Mueller et al., 2005*), a mouse antibody against GFP (Covance, Inc., Princeton, NJ), a rabbit antibody against *Chlamydomonas* D1bLIC (*Perrone et al., 2003*), and a rabbit antibody against mCherry (Rockland Immunochemicals, Limerick, PA).

### Drug treatment

0.34 mM $Ca^{2+}$ in TAP media was depleted by adding 0.5 mM EGTA, which resulted in a free $Ca^{2+}$ concentration of 1.5 μM. The concentration of free $Ca^{2+}$ in the assay buffer as a function of added EGTA was calculated from the Chelator program (http://maxchelator.stanford.edu). For dynein-1b inhibition assays, a final concentration of 100 μM ciliobrevin D was added to the TAP media, and the data was collected within 5–10 minof the treatment.

For cycloheximide treatment, cells were deflagellated by pH shock and cycloheximide was added to a final concentration of 1.5 μg/ml immediately afterwards. Cells were allowed to regrow flagella for 2 hr before fixation and imaging. For $Li^+$ treatment, 50 mM LiCl was added to liquid cell suspensions, and cells were incubated for 2 hr before fixation and imaging.

### Deflagellation and flagellar regrowth

For imaging the diffusion gradient in live *fla3::KAP-GFP* cells, we deflagellated cells in TAP media using shear force by rapidly pushing them through a 20G1 ½ syringe. Cells regenerating flagella were imaged in the following hour. For imaging the accumulation of GFP signal at the basal body region and in regenerating flagella, fla*3::KAP-GFP* and *IFT20::IFT20-GFP* cells were deflagellated with pH shock by adding 60 μl 0.5 N acetic acid to 1 ml of cells in TAP media, waiting 45 s, and adding 60 μl 0.5 N KOH. Cells were fixed 15, 30, 45, 60, and 75 min after pH shock. Fixation was done by pipetting 200 μl of liquid TAP cell culture onto a poly-lysine treated coverslip for 1 min, then gently treating the coverslip with 4% paraformaldehyde in water for 10 min. Afterwards, the coverslip was treated twice with 100% methanol chilled to −20 °C for 5 min. Coverslips were dipped in water to remove methanol, mounted in a flow chamber with TAP media, and then imaged immediately.

## TIR microscopy

A custom-built objective-type TIR fluorescence microscope was set up using a Nikon TiE inverted microscope equipped with a perfect focusing unit, bright-field illumination, and a 100-X 1.49 NA PlanApo oil immersion objective (Nikon, Melville, NY). 488 nm and 561 nm solid state lasers (Coherent, Santa Clara, CA) were used for GFP and crCherry excitation, respectively. The angle of incident light was adjusted lower than the critical angle to illuminate a deeper field (~300 nm) near the coverslip surface. The fluorescent signal was recorded by an iXon 512 $\times$ 512 electron-multiplied charge-coupled device (EM-CCD) camera (Andor, Belfast, United Kingdom). 1.5x extra magnification was used to obtain an effective pixel size of 106 nm. Data were collected at 10 Hz. Excitation laser beams were controlled by shutters (Uniblitz, Rochester, NY). Because the CCD image saturates under intense laser illumination of the focused gate beam, shutter timing was synchronized with the camera acquisition by a data acquisition card (NI, USB-6221) to minimize the number of saturated frames in recorded movies. For two-color imaging, GFP and crCherry fluorescence were separated into two channels on a CCD camera using Optosplit II (Cairn, Kent, United Kingdom). To avoid bleed-through between channels, movies were acquired using time-sharing between the 488 nm and 561 nm laser beams, synchronized with camera acquisition at 60 ms frame time. The effective pixel size was 160 nm.

## PhotoGate assays

The PhotoGate system was assembled as previously described (*Belyy et al., 2017*). Briefly, a 488 nm laser beam was split into two paths using a half-wave plate and a polarizer beamsplitter cube. The first path was used for objective-type TIR imaging. The second path was focused (2 MW cm$^{-2}$) to the image plane and steered with a fast piezo-driven mirror (S-330.8SL, Physik Instrumente, Karlsruhe, Germany). The piezo-driven mirror was mounted at a position conjugate to the back-focal plane of the objective to ensure that the tilting of the mirror resulted in pure translation of the focused beam in the image plane. The mirror provided a usable travel range of 30 μm x 30 μm area at the image plane. The mirror's angle was updated via analog output channels of a data acquisition card (National Instruments, Austin, TX, USB-6221) and controlled by software custom-written in LabVIEW.

Flagellar orientation of surface adhered cells was visualized by TIR imaging. Initially, the gate beam was placed at the tip of flagellum and moved along the flagellar orientation to prebleach the distal half of the flagellum. The gate beam was turned off when it was positioned near the base of the flagellum to allow a single fluorescent anterograde train to enter the flagellum. Occasionally (<5%), two anterograde trains overlapped and entered the flagellum simultaneously. The gate beam was then turned on for 0.2 s of every 1 s to bleach other anterograde trains. Based on the size of the gate beam in the image plane, gating frequency was adjusted such that less than 1% of anterograde IFT trains moved faster than the cutoff speed (3.0 μm s$^{-1}$). The trajectories of these trains can be distinguished from each other as they move at different speeds along the flagellum. The locations of flagellar tips were determined by brightfield imaging (data not shown). In two-color photogate experiments, the focused 488 nm laser beam was used to bleach both GFP and crCherry, and 488 and 561 beams were used in a time-sharing mode for TIR excitation.

## FRAP assays

FRAP assays on the *fla3::KAP-GFP* strain were performed by photobleaching the center part of the flagellum (5 μm in length) for 200 ms at 25 kW cm$^{-2}$ in the epifluorescence mode. The recovery of fluorescence signal in the bleached area was simultaneously monitored by imaging with a 100 W cm$^{-2}$ TIR excitation. The analysis was performed by measuring the total fluorescence intensity within the bleached area. Fluorescent signal of anterograde transport was manually excluded from the analysis. Thirteen different recovery traces were used in the MSD analysis. The intensity of each trace was normalized according to the initial and final intensity.

## Confocal microscopy

*fla3::KAP-GFP* and *IFT20::IFT20-GFP* cells were fixed with paraformaldehyde and imaged on a confocal microscope using 488 nm laser excitation (Zeiss, Oberkochen, Germany). Images were recorded with 560 nm *z* step, 63 nm pixel size, and 1.58 μs photon collection per pixel. Fluorescence in basal

body and flagellum was quantified using ImageJ. The ratio of flagellar to basal-body KAP-GFP fluorescence in confocal images was similar to that of live cell imaging under TIR excitation, indicating that the fixation protocol did not result in the loss of diffusing KAP-GFP signal from the flagella.

## Data analysis

Anterograde and retrograde trajectories were manually assigned from kymographs. After the arrival of a single anterograde particle at the tip, the departure of fluorescent retrograde trains was determined at single pixel and frame resolution. The tip resting time for each retrograde train was defined as the duration between the arrival of the fluorescent anterograde train and the departure of the retrograde train from the tip. Tip resting time histograms were constructed and fitted to a Gamma function using MATLAB. The Gamma function was defined as $\Gamma(t) = t^{\alpha-1}e^{-\lambda t}$, where $\alpha$ and $\lambda$ are shape and rate parameters, respectively.

For single particle tracking analysis, the positions of fluorescent spots were determined by a 2D Gaussian fitting algorithm. The positions were fitted throughout the movie except at the frames when the gate beam was on or the frames in which the tracked particle overlapped with other fluorophores. The intensity of the fluorescent spots was estimated by the volume of the 2D Gaussian peak. In a typical assay, we adjusted excitation power to achieve 20 nm localization accuracy at 10 Hz image acquisition rate. Individual GFP particles were tracked for 5 s on average before photobleaching and the diffusion constant was obtained by MSD analysis of individual spots. In certain kymographs, diffusion of individual KAPs within a flagellum could not be resolved due to the diffraction limit.

To determine the distribution of the KAP-GFP background in flagella, anterograde trajectories in kymographs of *fla3::KAP-GFP* cells were manually removed using custom ImageJ plugins. The remaining pixels were averaged over the kymograph's time axis, giving a time-averaged plot of the KAP-GFP background over the flagellum length. The cells were grouped by flagellar length. The background intensity and flagellum length of each cell were normalized. The average background intensity along the length of the flagellum was calculated for each group of cells.

KAP-GFP efflux from the flagellum was calculated using Fick's law. The slope of the KAP-GFP background over the length of a flagellum was multiplied by the diffusion constant (1.7 $\mu m^2$ $s^{-1}$). To calculate KAP-GFP influx, the KAP-GFP background was subtracted from the kymographs. Then, the average intensity of anterograde trains was multiplied by the train frequency (1.3 trains $s^{-1}$) to calculate the influx.

GFP photobleaching rate under TIR illumination was estimated by heavily decorating the coverslip surface with eGFP and calculating the rate of decrease in GFP fluorescence. To estimate the live-cell GFP photobleaching rate in *pf18 IFT27-GFP* cells, the fluorescent intensities of 94 anterograde trains from 9 cells were quantified at each time point en route to the flagellar tip. Each train's intensity profile was normalized by the mean intensity. The normalized intensity values were plotted against time and fit to a single exponential decay. The decay constant was used as the photobleaching rate.

## Monte Carlo simulations

Monte Carlo simulations were performed to test the effect of limited number of GFPs per train and GFP photobleaching in PhotoGate experiments using the *pf18 IFT27-GFP* strain. Experimentally measured values were used for the velocity and frequency of anterograde and retrograde trains. Simulations assumed that anterograde trains arrive and retrograde trains depart from the tip through a purely stochastic process, adding and subtracting particles to a mixed flagellar tip pool.

We estimated that each anterograde train contains six fluorescent GFPs by comparing the fluorescent intensities of anterograde trains in the *pf18 IFT27-GFP* strain to those of KAP-GFP spots in the *fla3::KAP-GFP* strain under the same imaging conditions and calibrating the number of molecules based on previous photobleaching analysis of the *fla3::KAP-GFP* strain (*Engel et al., 2009*). Each retrograde train was constructed by a random selection of IFT particles available at the tip. Tip intensity measurements revealed that the signal of the IFT complexes located at the tip is approximately three times brighter than an average anterograde train. The photobleaching of GFPs (0.07 $s^{-1}$) under TIR illumination was accounted for in simulations and trains with at least one fluorescent GFP upon leaving the tip were marked detectable.

Simulations were also run to estimate the distribution of diffusing KAP molecules in the flagellum at a steady-state. In these simulations, previously reported values for the anterograde train injection rate (1.3 trains s$^{-1}$) (*Mueller et al., 2005*) and the average number of KAP bound to a single anterograde train for each flagellar length (*Engel et al., 2009*) were used to estimate the number of KAP that arrives at the flagellar tip per second. KAP dissociated from the trains at the tip and immediately started one dimensional diffusion in the flagellum. The resting time of KAP at the tip was insignificant, and was not accounted for. The flagellum was modeled as a 5–12 μm long linear grid with spacing defined as the MSD of KAP diffusing at 1.7 μm$^2$ s$^{-1}$ (*Figure 3h*) during the time-step of the simulation (5 ms). At every time point, each active molecule had its grid position changed by +1 or −1. The molecules at the extreme terminus of the tip only moved towards the base. The diffusing KAP molecules were perfectly absorbed to the cell body as they arrived at the flagellar base (i.e. perfect sink) and exited the simulation. The simulations were run for 100,000 time points to allow molecules to reach a steady-state. The number of molecules at each grid position was calculated to plot the distribution of KAP molecules diffusing along the length of the flagellum. The total number of KAP was calculated by integrating the number of KAP diffusing along the entire flagellum and KAP on the anterograde trains. This number was compared to a hypothetical scenario that KAP returns to the cell body with active transport. The simulations were run 10 times to calculate the error.

Simulation codes are available on https://github.com/SingleMoleculeAC/IFT-Dynamics (*Chien, 2017*). A copy is archived at https://github.com/elifesciences-publications/IFT-Dynamics.

## Acknowledgements

We would like to thank K Augsperger for assistance with the isolation and screening of the *D1bLIC-Cherry KAP-GFP* tagged strains, V Belyy, J Bandaria, and P Qin for technical assistance, and W F Marshall and N L Hendel for sharing results prior to publication. This work has been supported by NIH (GM094522, GM116204 (AY), GM055667 (MEP)), and NSF (MCB-1055017 and MCB-1617028 (AY)).

## Additional information

### Funding

| Funder | Grant reference number | Author |
| --- | --- | --- |
| National Institute of General Medical Sciences | GM055667 | Mary E Porter |
| National Institute of General Medical Sciences | GM094522 | Ahmet Yildiz |
| National Science Foundation | MCB-1055017 | Ahmet Yildiz |
| National Science Foundation | MCB-1617028 | Ahmet Yildiz |
| National Institute of General Medical Sciences | GM116204 | Ahmet Yildiz |

The funders had no role in study design, data collection and interpretation, or the decision to submit the work for publication.

### Author contributions

Alexander Chien, Conceptualization, Data curation, Formal analysis, Investigation, Methodology, Writing—original draft, Writing—review and editing; Sheng Min Shih, Conceptualization, Data curation, Software, Formal analysis, Validation, Investigation, Writing—original draft; Raqual Bower, Data curation, Formal analysis, Validation, Investigation; Douglas Tritschler, Conceptualization, Data curation, Formal analysis; Mary E Porter, Conceptualization, Supervision, Writing—original draft; Ahmet Yildiz, Conceptualization, Supervision, Funding acquisition, Writing—original draft, Writing—review and editing

**Author ORCIDs**
Alexander Chien, http://orcid.org/0000-0003-1101-6721
Ahmet Yildiz, http://orcid.org/0000-0003-4792-174X

**Decision letter and Author response**
Decision letter https://doi.org/10.7554/eLife.28606.032
Author response https://doi.org/10.7554/eLife.28606.033

## Additional files

**Supplementary files**
• Transparent reporting form
DOI: https://doi.org/10.7554/eLife.28606.029

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
