## [Decision Letter]

Thank you for submitting your article "Dynamics of the IFT Machinery at the Ciliary Tip" for consideration by *eLife*. Your article has been reviewed by three peer reviewers, and the evaluation has been overseen by Anna Akhmanova as the Senior Editor and Reviewing Editor. The reviewers have opted to remain anonymous.

The reviewers have discussed the reviews with one another and the Reviewing Editor has drafted this decision to help you prepare a revised submission.

Summary:

In this manuscript, the authors adapt the recently developed PhotoGate method to track individual and small groups of IFT trains arriving at and departing from the ciliary tip in *Chlamydomonas*. The dynamics of the IFT machinery at the ciliary tip and the events driving train turnaround are key areas of interest in the field. Using the PhotoGate, the authors were able to visualize and quantify splitting of IFT trains with more clarity than previously possible. Their conclusions include that the train subunits mix at the tip, and that the anterograde motor kinesin-II returns to the base by diffusion, forming a system that helps control ciliary length. Overall, the reviewers found the approach and the questions asked to be very timely. However, they have raised a few substantive concerns regarding some conclusions and the manuscript presentation that would need to be addressed.

Essential revisions:

1) The results testing various agents to see if they affect the tip turnaround dynamics are interesting but raise many questions. For ciliobrevin, there is an increase in the return time of IFT27 and KAP. The authors should explain better what effect ciliobrevin has on anterograde and retrograde IFT and the number of these events. Furthermore, assuming that KAP dissociates from IFT complexes and returns to the cell body by diffusion, it is not clear why ciliobrevin affects its return time.

For IBMX and H-8, these appear to be very blunt tools. What kinases are affected? IBMX is probably better described as a phosphodiesterase inhibitor rather than a kinase (type unspecified) activator (subsection “IFT tip turnaround is regulated by dynein activity and extracellular Ca^2+^”, last paragraph), as it results in increased cAMP levels (which of course do increase PKA activity). H-8 is a kinase inhibitor that affects PKA, PKG and to some extent PKC and other kinases. The paper does not make clear how this relates to the statement about Ca^2+^ effects. Could the negative results with IBMX be because this reagent is not effective at this concentration in Chlamy? This is especially important since the literature has identified several kinases that directly affect IFT motors or at least affect IFT and cilia length, including kinases in the CDK, CaMK, and MOK families (for example, Tam et al. 2007, Liang et al. 2014, Burghoorn et al. 2007, Broekhuis et al. 2014). As a way of addressing this point, the authors could report how strongly other IFT parameters are affected, especially for H8. The key question that the authors need to address while revising this section is: do the treatments act directly on the machinery controlling return time, or reflect more general ciliary/IFT perturbation?

2) It is stated as fact that the IFT complexes mix at the tip (in the Abstract and elsewhere), but the data supporting these statements do not seem conclusive. This conclusion thus should be strengthened by additional data (see considerations below; a photoswitchable probe might be a useful approach). Alternatively, mixing should be mentioned as a plausible suggestion rather than a stated fact.

The main argument for mixing is that the number of retrograde trains departing the tip is not strictly proportional to the number of incoming anterograde trains. However, there seem to be caveats that prevent this from being the direct test desired:

a) Concurrent arrival of two trains. Given the relatively uniform velocity of anterograde IFT trains, it seems plausible that the PhotoGate will occasionally allow simultaneous entry of two trains that will remain in the same diffraction limited spot during transit to the tip, giving the appearance of one (bright) train. Inclusion of these events in the analysis could break strict proportionality without invoking mixing. The authors may be able to use simulation or analysis of train intensity to comment on the likelihood of this possibility. In general, a clear example of one anterograde train splitting into three or more retrograde trains is lacking from the manuscript.

b) Photobleaching. As the authors show, photobleaching has a large bearing on the observations (Figure 1) and accounting for the photobleaching rate brings the relationship between anterograde and retrograde trains closer to proportional. A photobleaching rate of 0.05/s is stated, but from which data was this key parameter determined and what is its confidence interval?

c) The finding that the average number of retrograde trains (2.4) per anterograde train is "significantly higher than the measured ratios of retrograde to anterograde train frequencies (~1.6) (Buisson et al., 2013; Dentler, 2005; Iomini et al., 2001; Mijalkovic et al., 2017)" is used as supporting evidence for mixing. However, the ratios in the cited papers are from a variety of species and differ widely (from 1 – 3.4). What is the ratio in the authors' experiments? In what sense is the difference "significant"; if statistically, what is the test and p-value?

3) A number of questions were raised regarding the proposed model for kinesin diffusion and flagella length control. At the very least, the questions below should be addressed by better discussion.

a) If the kinesin diffusion model for control of flagellar length is correct, that would presumably mean that mutants with long flagella (>20 um) have enhanced levels of kinesin compared to the strain used here. This seems an easy prediction to test.

b) It is not clear that the decrease in KAP at the basal body is really the limiting factor. The residual amount of KAP at the basal body likely is sufficient for some flagellar regrowth as *Chlamydomonas* can regrow approximately 1/2 length flagella after deflagellation in the absence of any additional protein synthesis (i.e. in cycloheximide). This can be tested in the KAP-GFP system.

c) Oligomeric status of kinesin-II during diffusion needs to be addressed more clearly. The authors nicely show that KAP departs the ciliary tip by diffusion in *Chlamydomonas*. An interesting observation is that the KAP spot (presumably consisting of multiple copies of the protein) does not split into smaller species, implying that Kap3 diffuses from the tip in the same oligomeric state as it arrives. This raises a number of questions:

- How many copies of KAP are present per train, based on calibrated fluorescence intensities (these are mentioned in methods, but I could not find the number in the manuscript)?

- What would hold KAP in oligomeric form if the train splits apart and mixes; do some components of the train remain intact?

- Why does the model of Figure 4 show single kinesin-II molecules diffusing from the tip, when this seems at odds with the data?

d) The authors claim that the anterograde transport and retrograde diffusion can explain the increasing fluorescence of KAP towards the tip. Do the other kinesin-2 subunits show a similar gradient along the flagellum? Is this seen for kinesin-2 motors in other systems?

4) Manuscript writing and organization.

a) The manuscript reads as a fusion of two pieces of work: one on IFT machinery dynamics at the tip (with its own discussion in subsection “Kinesin-2 carries dynein-2 as an inactive passenger during anterograde IFT”, third to last paragraphs –), and a second on kinesin-II diffusion and ciliary length control (which confusingly resumes with new results following the first discussion). Please combine all discussion in one place, and work on the cohesion of the two pieces.

b) The nomenclature that the authors use to describe events at the cilium tip is difficult to follow. In Figure 2, the authors label the figures as tip return time but the text refers to tip resting time, tip remodeling time, tip dwell time, and tip departure time. Please provide a schematic that describes each of these measurements as well as a definition in the text. Please be consistent between the figures and the text.

c) The differences among species and earlier work need to be discussed better.

- It needs to be more clearly stated that kinesin-II returns from the tip by active transport (rather than diffusion) in all metazoan cilia studied to date. Currently, this is buried in a confusing paragraph about kinesin-2 and Kif17/OSM-3. Similarly, in the last sentence of the Abstract it needs to be stated that dissociation of kinesin-II is so far specific to *Chlamydomonas* i.e. "facilitates flagellar length control in *Chlamydomonas*." The finding that diffusion of KAP from the tip is so far unique to *Chlamydomonas* is not a weakness of the work, showing the interesting diversity of the IFT system, and does not need to be hidden.

- In relation to earlier work, a larger decrease in KAP intensity at the flagellar base between short and full-length cilia is reported compared to Ludington et al. 2015. In addition, the authors do not find that the amount of IFT20-GFP in the flagellum is constant during regeneration, differing from Marshall and Rosenbaum's findings with IFT52, and a central feature of the balance point model. No explanation is offered for these differences.

d) The three paragraphs at the end of the section "Kinesin-2 carries dynein-2 as an inactive passenger during anterograde IFT" belong in the Discussion. In particular, the last of these paragraphs which discusses the differences between Chlamy, worm, and mammalian IFT processes, concerns one of the most important findings of the paper and needs to be prominently and thoroughly discussed.

---

## [Author Response]

Essential revisions:1) The results testing various agents to see if they affect the tip turnaround dynamics are interesting but raise many questions. For ciliobrevin, there is an increase in the return time of IFT27 and KAP. The authors should explain better what effect ciliobrevin has on anterograde and retrograde IFT and the number of these events.

We previously published the effect of ciliobrevin D in IFT movement in Shih et al. 2013 *eLife*. In that work “we varied ciliobrevin D concentration between 0–150 μM and monitored IFT 2 min after drug treatment in cells adhered both of their flagella to surface. We observed that both anterograde and retrograde IFT train frequencies dropped with increasing concentrations of ciliobrevin D, and at >100 μM ciliobrevin D we observed accumulation of IFT trains at the flagellar tip. At 150 μM ciliobrevin D, retrograde IFT frequency was reduced by 92%. The velocities of retrograde and anterograde trains also decreased by 60% and 36%, respectively.” See Figure 3 of Shih et al. for further information.

In this work, we used 100 μM ciliobrevin D, in which dynein motors are partially inactivated within 2-10 min of drug treatment. This concentration of ciliobrevin D results in 50% reduction in the frequency of retrograde and anterograde trains. It also leads to 50% and 28% reduction in retrograde and anterograde train velocities, respectively (Shih et al. 2013). We found that ciliobrevin D addition extends the tip resting time for a few seconds, suggesting that dynein activity is critical for the departure of IFT trains from the tip. The results are also consistent with tip accumulation of IFT trains and reduction in the frequency of retrograde trains in Shih et al. However, this delay in tip turnaround is not responsible for the observed reduction in train velocity and frequency, which must be related to the defects in retrograde transport.

We clarified the effect of ciliobrevin D in IFT motility in the main text by adding: “Addition of 0.1 mM ciliobrevin D to media results in 50% reduction in the frequency of retrograde and anterograde trains, and 50% and 28% reduction in retrograde and anterograde train velocities, respectively (56). In this case, the tip resting time of IFT27 increased over two-fold (Figure 2, p<10^−4^), suggesting that rapid turnover of IFT trains depends on dynein activity.”

Furthermore, assuming that KAP dissociates from IFT complexes and returns to the cell body by diffusion, it is not clear why ciliobrevin affects its return time.

In the revised version of our manuscript, we replaced “return time” with “tip resting time” to avoid confusion. We do not claim that ciliobrevin affects return of KAP to the base by diffusion. Instead, our results imply that ciliobrevin affects the time KAP remains at the tip before it starts diffusing in flagellum.

For IBMX and H-8, these appear to be very blunt tools. What kinases are affected? IBMX is probably better described as a phosphodiesterase inhibitor rather than a kinase (type unspecified) activator (subsection “IFT tip turnaround is regulated by dynein activity and extracellular Ca^2+^”, last paragraph), as it results in increased cAMP levels (which of course do increase PKA activity). H-8 is a kinase inhibitor that affects PKA, PKG and to some extent PKC and other kinases. The paper does not make clear how this relates to the statement about Ca^2+^ effects. Could the negative results with IBMX be because this reagent is not effective at this concentration in Chlamy? This is especially important since the literature has identified several kinases that directly affect IFT motors or at least affect IFT and cilia length, including kinases in the CDK, CaMK, and MOK families (for example, Tam et al. 2007, Liang et al. 2014, Burghoorn et al. 2007, Broekhuis et al. 2014). As a way of addressing this point, the authors could report how strongly other IFT parameters are affected, especially for H8. The key question that the authors need to address while revising this section is: do the treatments act directly on the machinery controlling return time, or reflect more general ciliary/IFT perturbation?

We did not observe any significant changes in IFT velocity and frequency in IBMX and H-8 treated cells (not shown). We agree with the reviewers’ concern that it is not clear which regulatory proteins are specifically affected with these treatments and whether these concentrations in extracellular medium are sufficient to achieve the desired effect in the first 5-20 minutes of treatment, which is the time period we collected our movies. Because of these shortcomings and the fact that these results are not critical for our conclusions, we removed the IBMX and H-8 data from the manuscript.

2) It is stated as fact that the IFT complexes mix at the tip (in the Abstract and elsewhere), but the data supporting these statements do not seem conclusive. This conclusion thus should be strengthened by additional data (see considerations below; a photoswitchable probe might be a useful approach). Alternatively, mixing should be mentioned as a plausible suggestion rather than a stated fact.The main argument for mixing is that the number of retrograde trains departing the tip is not strictly proportional to the number of incoming anterograde trains. However, there seem to be caveats that prevent this from being the direct test desired:a) Concurrent arrival of two trains. Given the relatively uniform velocity of anterograde IFT trains, it seems plausible that the PhotoGate will occasionally allow simultaneous entry of two trains that will remain in the same diffraction limited spot during transit to the tip, giving the appearance of one (bright) train. Inclusion of these events in the analysis could break strict proportionality without invoking mixing. The authors may be able to use simulation or analysis of train intensity to comment on the likelihood of this possibility.

The probability that two IFT trains overlap within a diffraction limited distance (~300 nm) along the length of a flagellum and their velocities are similar such that the distance between them does not increase over 300 nm during 10 µm travel along a flagellum is lower than 2%. Because the likelihood of such events is quite low, they do not affect the results in our simulations.

In general, a clear example of one anterograde train splitting into three or more retrograde trains is lacking from the manuscript.

We added an example kymograph to Figure 1—figure supplement 2 (see the left bottom corner of the figure) which shows one anterograde train to split into three retrograde trains.

b) Photobleaching. As the authors show, photobleaching has a large bearing on the observations (Figure 1) and accounting for the photobleaching rate brings the relationship between anterograde and retrograde trains closer to proportional. A photobleaching rate of 0.05/s is stated, but from which data was this key parameter determined and what is its confidence interval?

We calculated GFP photobleaching rate by immobilizing recombinantly expressed GFP at the coverslip surface and monitor the decrease of their fluorescence under our imaging conditions. Exponential fit of the signal over time revealed that the photobleaching rate is 0.05 s^-1^ in vitro (the confidence interval is below 0.005 s^-1^).

It is more challenging to measure GFP photobleaching rate in live *Chlamydomonas* cells, because multiple GFPs coexist on a single train. In addition, these trains move, sometimes overlap with each other, and they assemble/disassemble at the tip. To address the concern of the reviewers, we measured the decrease in fluorescence intensity of anterograde trains until they reach the tip. Exponential fit to this data revealed that the photobleaching rate is 0.07 ± 0.01 s^-1^ in cells (Author response image 1), similar to our calculation from the bulk in vitro assay mentioned in previous paragraph. Although this calculation method is more relevant to our measurements, it can monitor only the first 5 seconds of exposure before trains reach the tip, and is less reliable than the in vitro measurements. In either case, the time exponent in simulations is significantly lower than 1.

**Author response image 1. respfig1:** The normalized intensity of anterograde IFT20-GFP particles as they travel from the flagellar base (t = 0 s) to the tip. A single exponential decay fit (red curve) reveals a photobleaching rate of 0.07 ± 0.01 s^-1^ ( ± 95% c.i.).

c) The finding that the average number of retrograde trains (2.4) per anterograde train is "significantly higher than the measured ratios of retrograde to anterograde train frequencies (~1.6) (Buisson et al., 2013; Dentler, 2005; Iomini et al., 2001; Mijalkovic et al., 2017)" is used as supporting evidence for mixing. However, the ratios in the cited papers are from a variety of species and differ widely (from 1 – 3.4). What is the ratio in the authors' experiments? In what sense is the difference "significant"; if statistically, what is the test and p-value?

We found the retrograde-to-anterograde ratio to be 1.15, consistent with previous reports that retrograde frequencies are higher than anterograde trains. We ran a t-test to test the significance of the difference between retrograde-to-anterograde ratio and the number of retrograde trains per anterograde trains in PhotoGate experiments. The p-value is 10^-20^. We have clarified this section as follows:

“We directly observed that a single anterograde train splits into multiple retrograde trains at the tip (Figure 1, Figure 1—figure supplement 2). On average, 2.4 retrograde trains were detected departing from the tip after the arrival of a single fluorescent anterograde train (Figure 1 = 97), consistent with higher frequencies of retrograde IFT trains than anterograde IFT trains (Dentler, 2005; Iomini et al., 2001; Qin et al., 2007). However, the number of retrograde trains per fluorescent anterograde train in PhotoGate assays (2.4, Figure 1) was significantly higher (Welch’s t-test, p = 10^-20^) than the ratio of retrograde to anterograde train frequencies in conventional kymographs (1.15, Figure 1) (Dentler, 2005; Iomini et al., 2001; Reck et al., 2016). These observations suggested that IFT complexes from different anterograde trains recombine with each other to form retrograde trains at the tip.”

3) A number of questions were raised regarding the proposed model for kinesin diffusion and flagella length control. At the very least, the questions below should be addressed by better discussion.a) If the kinesin diffusion model for control of flagellar length is correct, that would presumably mean that mutants with long flagella (>20 um) have enhanced levels of kinesin compared to the strain used here. This seems an easy prediction to test.

The kinesin diffusion model states that kinesin-II available at the basal body becomes limiting as the flagella reach its full length. In this case, most KAP are trapped in flagella and their return is delayed by diffusion. There is also significant amount of kinesin-II available in the cytoplasm, even when cells fully grow their flagella. While the cytoplasmic pool of kinesin-II exchanges with the flagellar pool, the exchange rate is rather slow (unpublished observations) compared to the exchange between basal body and the flagellum. The model assumes that total amount of KAP localized to the flagellar compartment (basal body + flagellum) determines its length, while the cytoplasmic pool has no direct effect in length. Therefore, enhanced expression of kinesin-II would be one way to increase the equilibrium length, but this is not the only way. It is also possible to reach longer equilibrium length by reducing the flagellar disassembly rate, enhancing the recruitment of kinesin-II and other IFT components from the basal body to the flagella or affecting other pathways that are involved in flagellar length control. It is not clear by which specific mechanism lf mutations generate a long flagella phenotype.

We attempted the experiments the reviewers suggested by generating crosses of the KAP-GFP and lf strains to monitor KAP intensity in the lf background. In our preliminary observations, we found significant variability in individual cells, in terms of their flagellar length and KAP-GFP expression. In addition, IFT movement and tip turnaround behavior are significantly altered in some of the strains. Because of these complications, we feel that it will be difficult to compare KAP distribution in the lf background to the KAP-GFP cell line.

We have designed an alternative experiment to test the predictions of our model. It is previously shown that *Chlamydomonas* grows longer flagella in the presence of Li^+^. This occurs by recruiting flagellar proteins from the cell body pool into the flagella (Nakamura et al. 1987) rather than requiring new protein synthesis (Wilson et al. 2004), because addition of Li^+^ to cells that already grew their flagella leads to a 50% increase in flagellar length even in the presence of the protein synthesis inhibitor cycloheximide. Based on these results, addition of Li^+^ is expected to increase available KAP for flagella to 1.5X compared to control cells. Because these assays can be performed using the same strain, they are more likely to yield interpretable observations.

We grew the KAP-GFP strain in 50 mM Li^+^ and in regular TAP media. In addition, we deflagellated cells and regrew their flagella in the presence of cycloheximide. Consistent with previous observations, cells grew their flagella to 1.5X of their normal length in Li^+^. After these cells reached their steady state length, we calculated the total KAP intensity at the basal body and the flagellum. Consistent with our model, we observed that KAP gets depleted at the basal body at equilibrium. The KAP fluorescence localized to a flagellum correlated strongly with flagellar length, similar to untreated cells. The total KAP fluorescence was 50% higher in Li^+^ treated cells than untreated cells. These experiments support the predictions of our model that altering the amount of KAP available for the flagellar compartment positively correlates with the flagellar length and that the equilibrium length is set when KAP gets depleted below a certain threshold at the basal body.

In the revised manuscript, we showed these results in Figure 6 and Figure 6—figure supplement 1. We also extensively explained these experiments in Results, Discussion and Materials and methods.

b) It is not clear that the decrease in KAP at the basal body is really the limiting factor. The residual amount of KAP at the basal body likely is sufficient for some flagellar regrowth as *Chlamydomonas* can regrow approximately 1/2 length flagella after deflagellation in the absence of any additional protein synthesis (i.e. in cycloheximide). This can be tested in the KAP-GFP system.

KAP intensity localized to the flagellum after regrowth is several fold higher than KAP localized to the basal body in a steady state. Therefore, the majority of the material required for regrowth is supplemented from the cytoplasmic pool, not from the residual amount left in the basal body.

In the presence of cycloheximide, *Chlamydomonas* can grow half-length flagella after deflagellation, suggesting that the cytoplasmic pool of flagellar proteins is at least one half of that localized to the flagellar compartment. This material can be effectively recruited to the basal body after deflagellation even in the absence of new protein synthesis. This treatment is expected to decrease available KAP to 0.5X for the flagellar compartment. Consistent with previous observations, cells grew their flagella to 0.5X of their normal length in cycloheximide. In a steady state, we observed that KAP gets depleted at the basal body at equilibrium. The KAP fluorescence localized to a flagellum correlated strongly with flagellar length similar to untreated cells. The total KAP fluorescence was 50% lower in cycloheximide treated cells than untreated cells.

In the revised manuscript, we showed these results in Figure 6 and Figure 6—figure supplement 1. We also extensively explained these experiments in Results, Discussion and Materials and methods.

c) Oligomeric status of kinesin-II during diffusion needs to be addressed more clearly. The authors nicely show that KAP departs the ciliary tip by diffusion in *Chlamydomonas*. An interesting observation is that the KAP spot (presumably consisting of multiple copies of the protein) does not split into smaller species, implying that Kap3 diffuses from the tip in the same oligomeric state as it arrives. This raises a number of questions:

We are also surprised by this observation. We have strong evidence that KAP molecules from a single anterograde train depart from the tip simultaneously (i.e. within a 100 ms frame time), because in most cases the KAP intensity at the tip decreases to background levels in a single frame. After departure, KAP starts rapidly diffusing in the flagellum. In some cases, we observe a single fluorescent spot to diffuse for at least 5-10 s after departure. In other cases, the KAP-GFP signal spreads up quickly along the length of a flagellum, presumably because KAPs split apart and diffuse alone. In the latter case, it is not possible for us to resolve individual molecules because of diffraction-limited resolution (250 nm) of the tracking approach, low fluorescent signal, and limited time resolution (0.1 s) compared to rapid diffusion of the molecules (1.7 µm^[2]^/s). Therefore, we have strong evidence to suggest that KAP departs together from the tip and in certain cases they diffuse together at the first few seconds after departure. It is also likely that KAP diffuses individually in other cases.

To highlight this second possibility, we added “Unlike IFT trains, the majority (89%, N = 95) of KAP-GFPs simultaneously departed from the tip in a single step (Figure 3—figure supplement 1). […] The rest of the kymographs were ambiguous.”

- How many copies of KAP are present per train, based on calibrated fluorescence intensities (these are mentioned in methods, but I could not find the number in the manuscript)?

Based on the photobleaching analysis (Engel et al. 2009 JCB), we anticipate that there are 6 fluorescent GFPs on each anterograde KAP particles. We note that this is the lower boundary estimate for the number of KAP subunits per train because significant fraction of GFPs remain non-fluorescent in physiological conditions (Ulbrich et al. Nat. Methods 4, 319–321 (2007)). In addition, some GFPs may photobleach before we start recording movies or may not be detected.

- What would hold KAP in oligomeric form if the train splits apart and mixes; do some components of the train remain intact?

We do not know whether KAP oligomerizes on its own or some components of the train also dissociate and oligomerize with KAP. We feel it would be too speculative to comment on this point in the manuscript because observed oligomerization may be short lived and occurs in only a subset of the traces, while in other cases KAP may diffuse individually.

In the Discussion, we added: “In certain cases, KAP-GFP appears to diffuse in an oligomeric form. It remains to be studied what holds KAPs together and whether other components of the IFT trains diffuse with KAP clusters after splitting and mixing at the tip.”

- Why does the model of Figure 4 show single kinesin-II molecules diffusing from the tip, when this seems at odds with the data?

We showed KAP diffusing as a cluster and individually in the revised model.

d) The authors claim that the anterograde transport and retrograde diffusion can explain the increasing fluorescence of KAP towards the tip. Do the other kinesin-2 subunits show a similar gradient along the flagellum? Is this seen for kinesin-2 motors in other systems?

We do not have direct ways to test whether other subunits show a similar gradient along the flagellum, because *Chlamydomonas* strains with fluorescent fusions to these subunits do not exist. However, several observations suggest that other subunits also return to the base without a need for retrograde IFT. In Discussion, we wrote: “We propose that the diffusion of KAP-GFP represents the movement of the entire heterotrimeric kinesin-II complex because KAP and the kinesin-II motor subunits co-sediment in sucrose density gradients of purified flagella extracts (Cole et al., 1998; Mueller et al., 2005) and neither KAP nor FLA10 accumulate in flagella during inactivation of retrograde transport (Engel et al., 2012; Pedersen et al., 2006; Reck et al., 2016).”

If only KAP were to dissociate from IFT but the heavy chains return to the base with retrograde IFT, FLA10 would be expected to remain in flagellum after inactivation of retrograde IFT.

4) Manuscript writing and organization.a) The manuscript reads as a fusion of two pieces of work: one on IFT machinery dynamics at the tip (with its own discussion in subsection “Kinesin-2 carries dynein-2 as an inactive passenger during anterograde IFT”, third to last paragraphs), and a second on kinesin-II diffusion and ciliary length control (which confusingly resumes with new results following the first discussion). Please combine all discussion in one place, and work on the cohesion of the two pieces.

We combined these two sections under Discussion as requested.

b) The nomenclature that the authors use to describe events at the cilium tip is difficult to follow. In Figure 2, the authors label the figures as tip return time but the text refers to tip resting time, tip remodeling time, tip dwell time, and tip departure time. Please provide a schematic that describes each of these measurements as well as a definition in the text. Please be consistent between the figures and the text.

We apologize for this confusion. As reviewers suggested, we defined each parameter in a schematic in new Figure 2. We also carefully defined each parameter mentioned in the text consistent with the figures.

c) The differences among species and earlier work need to be discussed better.- It needs to be more clearly stated that kinesin-II returns from the tip by active transport (rather than diffusion) in all metazoan cilia studied to date. Currently, this is buried in a confusing paragraph about kinesin-2 and Kif17/OSM-3.

We added the following sentence to Discussion: “Unlike *Chlamydomonas*, kinesin-II returns from the tip by active transport, rather than diffusion, in all metazoan cilia studied to date.”

Similarly, in the last sentence of the Abstract it needs to be stated that dissociation of kinesin-II is so far specific to *Chlamydomonas* i.e. "facilitates flagellar length control in *Chlamydomonas*." The finding that diffusion of KAP from the tip is so far unique to *Chlamydomonas* is not a weakness of the work, showing the interesting diversity of the IFT system, and does not need to be hidden.

We made this change in the Abstract as suggested.

- In relation to earlier work, a larger decrease in KAP intensity at the flagellar base between short and full-length cilia is reported compared to Ludington et al. 2015.

Our results largely agree with Ludington et al. 2015, except we see a ~4-fold decrease while Ludington et al. reported a ~2-fold decrease in KAP intensity at the base. Although the underlying reason for this difference remains unclear, it may be related to different procedures used in fixation, parameters used in confocal imaging or background correction.

In addition, the authors do not find that the amount of IFT20-GFP in the flagellum is constant during regeneration, differing from Marshall and Rosenbaum's findings with IFT52, and a central feature of the balance point model. No explanation is offered for these differences.

We also noticed that this result contradicts with W. Marshall’s Western Blot analysis of IFT139 and IFT81, and immunolabeling results of IFT52. To ensure that we see an increase in IFT20-GFP accumulation in flagellum during regrowth, we have repeated our measurements (See Author response image 2). We have obtained results similar to what we present in our manuscript. It is not clear what causes these differences between our image analysis versus Western Blot or immunolabeling of flagella. We image a different IFT component and use a different methodology for fixation, data analysis and background correction in comparison to W. Marshall’s studies. In either case, the key point of this analysis is to show that amount of IFT component localized to the basal body remains nearly constant during flagellar regrowth and hence it is unlikely that the number of IFT components at the basal body are critical for limiting the flagellar elongation. This is in stark contrast to KAP-GFP, which is gradually depleted at the basal body during regrowth and accumulates more heavily in flagellum than IFT20 in a fully-grown flagellum.

We added “This discrepancy may be related to differences in methods for quantifying IFT components in flagella.”

A central feature of the balance point model is that at least one component of the IFT machinery becomes limited as the flagella elongate and this leads to the reduction of the flagellar assembly rate. When the assembly rate becomes as low as the length-independent disassembly rate, elongation stops and equilibrium length is reached. The model would be void if there is an infinite pool of components available in the cytoplasm and this pool maintains the number of particles at the basal body. Our model is consistent with the postulations of the balance-point model and shows that KAP gets depleted at the basal body, not maintained, as the flagella elongate. Therefore, our results explain which IFT component becomes limited as flagella elongate and why this specific component gets depleted instead of others (i.e. diffusion).

**Author response image 2. respfig2:** Integrated IFT20-GFP fluorescence at the basal body (left), in the flagellum (middle), and in both regions (right) at different flagellar lengths. Each black dot represents a single flagellum and the blue line is the running average ( ± s.e.m., N = 214 flagella from 107 cells).

d) The three paragraphs at the end of the section "Kinesin-2 carries dynein-2 as an inactive passenger during anterograde IFT" belong in the Discussion.

We moved this section to Discussion.

In particular, the last of these paragraphs which discusses the differences between Chlamy, worm, and mammalian IFT processes, concerns one of the most important findings of the paper and needs to be prominently and thoroughly discussed.

We expanded this section to clearly identify differences between IFT and ciliary length control in *Chlamydomonas* in Discussion.